# Polymer-coated carbon nanotube hybrids with functional peptides for gene delivery into plant mitochondria

Simon Sau Yin Law [1], Geoffrey Liou [1], Yukiko Nagai[2], Joan Giménez-Dejoz[1], Ayaka Tateishi[3], Kousuke Tsuchiya [1,3], Yutaka Kodama [1,4], Tsuyohiko Fujigaya[2] & Keiji Numata [1,3 ✉]

The delivery of genetic material into plants has been historically challenging due to the cell wall barrier, which blocks the passage of many biomolecules. Carbon nanotube-based delivery has emerged as a promising solution to this problem and has been shown to effectively deliver DNA and RNA into intact plants. Mitochondria are important targets due to their influence on agronomic traits, but delivery into this organelle has been limited to low efficiencies, restricting their potential in genetic engineering. This work describes the use of a carbon nanotube-polymer hybrid modified with functional peptides to deliver DNA into intact plant mitochondria with almost 30 times higher efficiency than existing methods. Genetic integration of a folate pathway gene in the mitochondria displays enhanced plant growth rates, suggesting its applications in metabolic engineering and the establishment of stable transformation in mitochondrial genomes. Furthermore, the flexibility of the polymer layer will also allow for the conjugation of other peptides and cargo targeting other organelles for broad applications in plant bioengineering.

[1] Biomacromolecules Research Team, RIKEN Center for Sustainable Resource Science, Wako, Saitama 351-0198, Japan. [2] Department of Applied Chemistry, Kyushu University, Nishi-ku, Fukuoka 819-0395, Japan. [3] Department of Material Chemistry, Kyoto University, Nishikyo-ku, Kyoto 615-8510, Japan. [4] Center for Bioscience Research and Education, Utsunomiya University, Utsunomiya, Tochigi 321-8505, Japan. ✉email: numata.keiji.3n@kyoto-u.ac.jp

Plant biotechnology remains one of the most promising approaches for scaling food and energy production to meet the growing demands of the world population[1,2]. This has expanded into engineering plants as biofactories capable of producing useful chemicals, including biopharmaceuticals, recombinant proteins, and fuel[3,4]. In particular, mitochondria are important targets due to their influence on agronomic traits such as fertility, plant vigor, and cross-compatibility[5]. A robust mitochondrial delivery system could be used to engineer traits such as cytoplasmic male sterility into new plant species and shorten breeding programs[6,7]. Furthermore, the role of mitochondria in energy generation and respiratory pathways also produces unique metabolites and compounds with applications in therapeutics and natural product synthesis[8,9].

Despite these potential applications, plant mitochondria have remained elusive for genetic engineering. Delivered cargo must first pass through the exterior plant cell wall, which has a small (5–20 nm) size exclusion limit, followed by the inner mitochondrial membrane, which has low permeability to hydrophilic molecules[10,11]. This is further compounded by the lack of selectable markers for mitochondrial transformation, making it difficult to screen for successful delivery. Our previous studies demonstrated that plasmid DNA (pDNA) complexed with a peptide containing the polycationic sequence (nine lysine-histidine repeats [KH9]) and the mitochondria targeting sequence (the presequence of the mitochondrial cytochrome $c$ oxidase subunit IV [Cytcox]) was capable of delivering DNA into mitochondria[12,13].

Carbon nanotubes (CNTs) are cylindrical-tubule structures made from graphite with exceptional physical properties that have been harnessed in biological applications such as drug delivery and biosensors[14,15]. They are capable of crossing plant cell walls and membranes due to their high aspect ratio, surface area, and stiffness and have emerged as potential solutions to alleviate plant delivery challenges[16–18]. CNT-based nanocarriers have been shown to efficiently deliver nucleic acid cargo into intact plants, including chloroplasts, without significant cytotoxicity[2,19]. However, they have yet to be demonstrated as viable vehicles for delivery into plant mitochondria.

Existing applications often involve the direct covalent modification of CNTs to target and bind different cargoes for delivery[20]. Direct modification of CNTs has low yields and substantially affects their physical and optical properties, which can attenuate their function as trackable probes[21–23]. To address these shortcomings, we applied our previously developed single-walled carbon nanotube (SWNT) polymer hybrids with a thiol-reactive maleimide methacrylate layer that allows for highly flexible functionalization as a CNT-based delivery system[24–26]. The maleimide methacrylate layer is non-covalently adsorbed on the surface of the SWNT using micelle polymerization and can be covalently conjugated with thiol-containing moieties. The Cytcox peptide contains the mitochondrial targeting presequence that has been previously characterized and used for mitochondrial pDNA delivery and expression within plant and animal cells[26]. KH9 is a cationic peptide that facilitates electrostatic interactions with anionic pDNA for binding.

In this work, we combine the cell penetrating abilities of CNTs with the targeting and DNA binding properties of Cytcox and KH9 peptides to develop a robust method of DNA delivery into the mitochondria of intact *Arabidopsis thaliana* (*A. thaliana*) (Fig. 1). Compared with previous peptide-only approaches, this SWNT/polymer hybrid achieve a near 30-fold increase in expression without toxicity effects[13]. In addition, we show that the delivered DNA is capable of homologous recombination within the mitochondrial genome. Our results also indicate that the expression of a folate metabolism-related gene confers increased root growth and folate levels with potential for metabolic engineering, including the development of a selection marker system for stable transformation of plant mitochondria. The flexibility of this platform will allow further development to deliver other biomolecules targeting different organelles and suggests that it can have broad potential applications in plant bioengineering.

## Results and discussion

**Preparation and characterization of SWNT nanocarriers.** SWNTs coated with a cross-linked polymethacrylate maleimide polymer network (SWNT-PM) were prepared using our previously developed micelle polymerization method as outlined in Supplementary Fig. 1[24,25]. Furan-protected maleimide was polymerized with polyethylene glycol (PEG) methacrylate as a non-covalently bound polymer layer on dispersed SWNTs (Supplementary Fig. 1a). Upon deprotection, the maleimide group can be covalently modified with thiol-containing molecules via Michael addition (Supplementary Fig. 1b, c). To target SWNT-PM for DNA delivery into mitochondria, we covalently conjugated two distinct Cys-containing functional peptides to yield SWNT-PM-CytKH9 (Fig. 1a). CytcoxCys (MLSLRQSIRFFKC), abbreviated as Cyt in SWNT-PM-CytKH9, has been previously shown to direct the yeast cytochrome $c$ oxidase subunit IV into the mitochondrial matrix and deliver pDNA for transient expression into mitochondria within plant and animal cells[26]. KH9 (KHKHKHKHKHKHKHKHKHC) is a cationic peptide consisting of alternating lysine and histidine residues that facilitates electrostatic complexation with anionic pDNA to increase loading efficiency. Our previous studies using these peptides have shown that they are able to direct nucleic acid and protein cargoes to the mitochondria within intact plants[12,13].

The SWNT nanocarriers (NCs) were quantitatively and qualitatively evaluated for their physical and chemical properties. Field-emission scanning electron microscopy (FE-SEM) of SWNT-PM and SWNT-PM-CytKH9 (6 kV on Si) (Fig. 2a, b) show the typical bundled morphology of SWNTs approximately 200 nm to 2 μm in length, suggesting that the adsorption of the polymer layer and conjugation of the peptides did not significantly alter the physical dimensions of the SWNTs. Similar bundles could be observed in the SWNT-PM-CytKH9 sample with pDNA (Fig. 2c), which showed round condensed pDNA particles that resemble those shown in previous studies of CNT and DNA complexes[27–29].

Thermogravimetric analysis (TGA) of the SWNT NCs was used to compare the mass loss profiles from 30 to 500 °C (Supplementary Fig. 2) and evaluate their composition. Pristine SWNTs had minimal weight loss during pyrolysis, as expected for CNTs. SWNT-PM and SWNT-PM-CytKH9 showed a large mass loss at approximately 300 °C, which was completed at approximately 420 °C. This agrees well with residual amounts after pyrolysis of the maleimide and peptide samples (Supplementary Fig. 2) and suggests that it corresponds to the adsorbed polymer layer and conjugated peptides. The residual weights at 500 °C for SWNT-PM (22%) and SWNT-PM-CytKH9 (33%) suggested that the weight ratio of polymer to SWNT to peptide is approximately 10:3:2 in SWNT-PM-CytKH9.

Raman spectroscopy of the SWNT NCs in solution also confirmed that the typical characteristic Raman peaks for SWNT were still detected and that no significant physical structural changes were introduced to the SWNT. Prominent G-bands at 1590 cm$^{-1}$ and G′-bands at 2640 cm$^{-1}$ could be observed despite the polymer coating (Supplementary Fig. 3). The successful conjugation of the peptide and formation of the polymethacrylate layer were also confirmed by using alkaline hydrolysis to cleave and release the PEG-methacrylate conjugated peptides. The

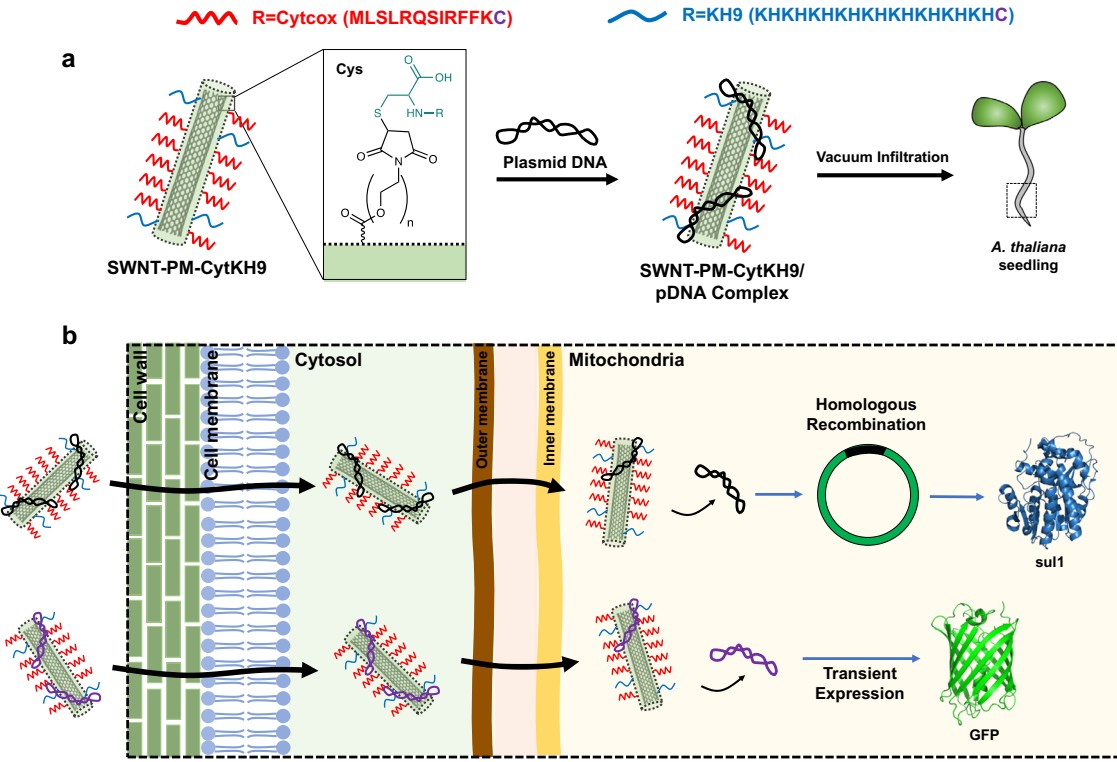

**Fig. 1 Overall schematic of the SWNT-PM-CytKH9 delivery system. a** Mitochondria-targeting (Cytcox in red) and DNA-complexing (KH9 in blue) peptides with a Cys on the C-terminus are functionalized (inset) via Michael addition to the polymer layer (green) on the SWNT. The resulting SWNT-PM-CytKH9 is complexed with pDNA and introduced into *A. thaliana* via vacuum infiltration. Both peptides contain an extra Cys residue at the C-terminus for conjugation to the maleimide group on the polymer. **b** SWNT-PM-CytKH9 passes through the cell and mitochondrial membranes to deliver and release the pDNA cargo in the mitochondria. Depending on the pDNA used, the pDNA can be transiently expressed or undergo homologous recombination before expression.

released fragments were detected via high-performance liquid-phase chromatography (HPLC) and mass spectrometry (MALDI-TOF/MS) (Fig. 2d and Supplementary Fig. 4). HPLC chromatograms of the PEG-methacrylate conjugated peptides cleaved from the SWNT-PM-CytKH9 sample showed two sets of distinct peaks corresponding to the respective peptides (Supplementary Fig. 4a).

Based on the integrated peak intensities from the mass spectra of the samples (Fig. 2d and Supplementary Fig. 4b–d), three distinct regions (orange, blue, and green) in the *m/z* spectra could be detected, which corresponded to the unfunctionalized, Cytcox, and KH9 polymer fragments, respectively. The size distribution between the peaks in each region ($m/z = 44$) corresponded to the repeating units of PEG in the functionalized methacrylate groups upon cleavage and further provide evidence of successful conjugation with the respective peptides.

**DNA binding ability of SWNT-PM-peptide**. We then evaluated the ability of each SWNT-PM-Peptide and its ability to form a stable complex with pDNA (*pDONR-35S-GFP*) used in subsequent experiments and the effects of the DNA binding peptide KH9. SWNT-PM-Cytcox and SWNT-PM-KH9 containing only each respective peptide were also prepared, and pDNA readily complexed with all SWNT-PM-Peptide forms. All samples followed a typical single-substrate binding curve, as characterized by electrophoresis agarose gel shift assays (Fig. 2e and Supplementary Fig. 5). Quantification of the corresponding free pDNA band (~4000 bp) at various SWNT-PM-Peptide amounts revealed that SWNT-PM-KH9 was most effective at complexing pDNA and required 1 μg to bind 50 ng of pDNA (Fig. 2f). SWNT-PM-CytKH9 and SWNT-PM-Cytcox required 25 and 100 μg to bind the same 50 ng of pDNA, respectively, and showed that SWNT-

PM-CytKH9 and SWNT-PM-KH9 were 4 and 30 times more effective at complexing DNA than SWNT-PM-Cytcox. This difference in DNA binding may be explained in part by the measured surface charge zeta potentials of the SWNT-PM-Peptide (Fig. 2g). SWNT-PM-Cytcox had a more negative zeta potential (−34.1 mV) than SWNT-PM-CytKH9 (18.2 mV), and SWNT-PM-KH9 (30.8 mV) and positively charged surfaces were likely more favorable for binding with negatively charged pDNA. This was also reflected in the change in zeta potential upon complexation with pDNA, where SWNT-PM-KH9 saw the greatest change (−22.2 mV) when compared with SWNT-PM-CytKH9 (−4.8 mV), and SWNT-PM-Cytcox (+2.4 mV).

Atomic force microscopy (AFM) was then used to characterize the polymer layer and DNA binding (Fig. 2h–j). Representative AFM images of SWNT-PM and SWNT-PM-CytKH9 showed that the overall morphology and lengths of the SWNTs were unchanged upon polymer coating and peptide functionalization (Fig. 2h, i). Long strands of SWNT-PM could be observed on the graphite substrate and on average ranged from 200–1000 nm that corresponded well with the average lengths (200–400 nm and PDI: 0.4–0.55) observed by dynamic light scattering (DLS) (Supplementary Fig. 6). Upon pDNA complexation, however, short strands of pDNA could be seen on the surface of the coated SWNTs, similar to those observed in the FE-SEM results (Fig. 2i). The profiled AFM heights were relatively uniform for the measured samples (Fig. 2j) and increased from 0.81 ± 0.26 nm to 2.14 ± 0.24 nm upon polymer coating, giving a polymer layer thickness of approximately 1.3 nm, which agrees with our previous results[24]. Functionalization with the Cytcox and KH9 peptides did not significantly change the observed heights, which could be due to the AFM measurement conditions and the overall flexibility of

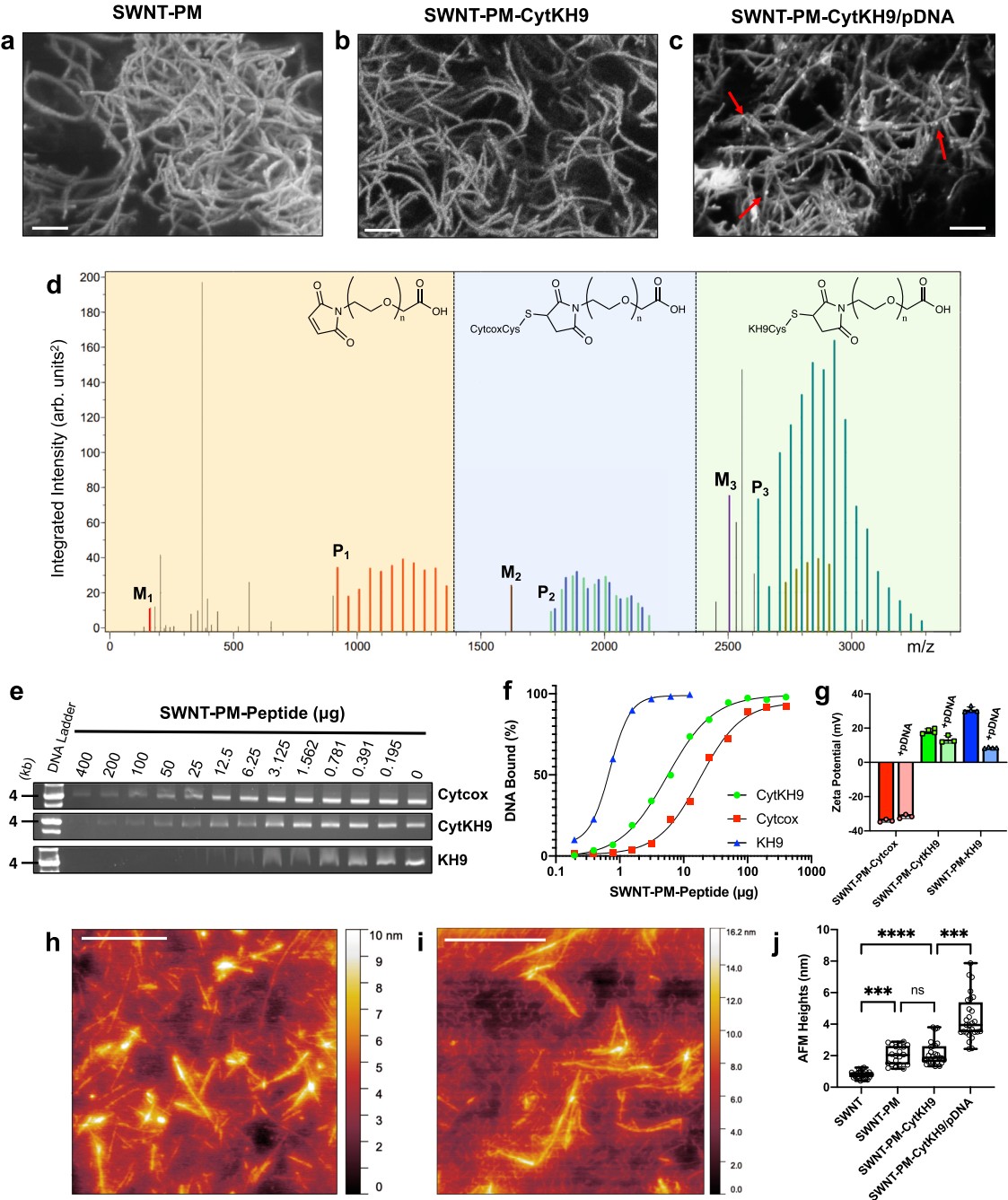

the peptides that limits their height contribution compared to the polymer layer. Complexation of pDNA on the SWNT NCs further increased the observed heights ($4.36 \pm 0.24$ nm), suggesting that there were sufficiently strong interactions between the pDNA and SWNT NCs to form a stable complex.

We also performed molecular dynamics (MD) simulations using polymer parameters based on the earlier TGA analysis. A system of three 50-unit polymer molecules on a 20 nm SWNT molecule (1 nm diameter) was constructed in an explicit water environment (Supplementary Fig. 7). Analysis of the cross-sectional diameters of the polymer-adsorbed SWNTs showed an average of 3.2 nm through mostly electrostatic interactions and achieved equilibrium after 150 ns (Supplementary Fig. 7b, c). This agrees well with our AFM results, suggesting that there is a fairly uniform polymer layer.

In summary, the physical characterization of the SWNT NCs demonstrated that the overall unique physical properties of the

SWNT are unchanged upon polymer adsorption and peptide conjugation, suggesting that this could be a flexible platform that can be modified for different delivery applications. pDNA was also shown to complex readily with the SWNT-PM-Peptide, and the incorporation of KH9 significantly improved the amount bound and should likely translate into a greater amount of DNA delivered.

**Uptake of SWNT-PM-Peptide/pDNA within A. thaliana.** To investigate the delivery capabilities of the SWNT NCs to plant mitochondria, we prepared fluorescently-labeled SWNT-PM-CytKH9 using DyLight488-conjugated KH9Cys and examined its localization within the root cells of *A. thaliana* upon vacuum/ pressure infiltration at 0.08 MPa for 60 s. In the cells infiltrated with the labeled SWNT-PM-CytKH9, clear colocalization between DyLight488-labeled SWNT-PM-CytKH9 and MitoTracker-stained

**Fig. 2 Physical and chemical characterization of SWNT NCs and their DNA binding characteristics.** FE-SEM micrographs of (**a**) SWNT-PM, (**b**) SWNT-PM-CytKH9, and (**c**) SWNT-PM-CytKH9/pDNA show typical SWNT morphology after excess peptide has been removed by dialysis. Scale bars represent 200 nm. White arrows indicate complexed pDNA on the surface of the SWNT. Representative micrographs are shown from 4 imaged drops for each condition ($n = 4$). **d** Polymer analysis using the MALDI-TOF/MS data of the peptide fragments cleaved from SWNT-PM-CytKH9 upon alkaline hydrolysis shows the size distribution of the incorporated PEG for unfunctionalized polymer (orange), Cytcox-functionalized polymer (blue), and KH9-functionalized polymer (green). The single maleimide methacrylate monomer unit and unconjugated Cytcox and KH9 peptides are labeled $M_1$, $M_2$, and $M_3$, respectively, and the lowest identifiable peaks for the conjugated species are labeled $P_1$, $P_2$, and $P_3$ and represent a single methacrylate monomer unit conjugated to the respective peptides. Pairs of peaks in regions 2 and 3 correspond to the oxidized ions ($m/z + 16$). **e** Representative gel shift electrophoresis mobility assay showing the amount of residual uncomplexed DNA at different SWNT-PM-Peptide/pDNA ratios. **f** Quantification of the respective band intensities from (**e**) to estimate DNA bound for each SWNT-PM-Peptide. **g** Zeta potentials of SWNT-PM-Peptide with different functional peptides in the absence and presence of pDNA. Data are represented as the mean ± standard deviation values ($n = 3$). **h, i** AFM micrographs of SWNT-PM-CytKH9 in the (**h**) absence and (**i**) presence of pDNA on a graphite surface. Scale bars represent 500 nm. Representative AFM images are shown from three measured samples for each sample condition ($n = 3$). **j** Quantified AFM heights of pristine SWNT and SWNT NCs. Cross-sectional topography measurements ($n = 25$) were taken to estimate the respective heights across a minimum of three AFM samples for each condition. Data points are individually shown with the respective box and whisker plots depicting the maxima, minima, median and the first (25%) and third (75%) quartiles; Significance between individual samples was determined by one-way ANOVA. $P$-values are 0.0002 for comparisons between SWNT and SWNT-PM, 0.0002 between SWNT-PM-CytKH9 and SWNT-PM-CytKH9/pDNA, and <0.0001 between SWNT and SWNT-PM-CytKH9. ns – not statistically significant, ***$P < 0.001$, ****$P < 0.0001$. Source data are provided as a Source Data file.

mitochondria could be observed in the samples infiltrated with the labeled SWNT-PM-CytKH9 as well as localization of the labeled SWNT-PM-CytKH9 within the cytosol of the root cells (Fig. 3a, upper panels). Conversely, most of the labeled SWNT-PM-CytKH9 samples remained on the surface of the root without vacuum infiltration (Fig. 3a, middle panels) and minimal fluorescence was observed for seedlings infiltrated with the free DyLight488-labeled KH9 (Fig. 3a, lower panels).

Next, we analyzed isolated mitochondria from *A. thaliana* infiltrated with SWNT NCs by confocal Raman microscopy (Fig. 3b–d). Raman mapping of the G band peak at 1590 cm$^{-1}$ showed clear colocalization of SWNT signal with the isolated mitochondria from seedlings infiltrated with SWNT-PM-KH9 and SWNT-PM-CytKH9 (Fig. 3b) that is not present in the samples infiltrated with SWNT-PM. Similar levels of colocalization were also observed in samples infiltrated with SWNT-PM-CytKH9 but those infiltrated with SWNT-PM-KH9 and SWNT-PM exhibited considerably less G-band signal, suggesting that the Cytcox peptide played a role in directing the SWNT NCs into the mitochondria.

To further quantify the effect of the peptide conjugation, we collected Raman spectra over an area of isolated mitochondria treated with the SWNT NCs and the overlay of the averaged Raman spectra (Fig. 3c) showed similar characteristic G and G′ bands at 1590 cm$^{-1}$ and 2640 cm$^{-1}$ (Fig. 3c, d). Quantification of the normalized 1590 cm$^{-1}$ (Fig. 3d) and 2640 cm$^{-1}$ (Supplementary Fig. 8) peak heights showed that all three SWNT NCs were detected within the isolated mitochondria. In particular, the strongest Raman signals were detected from the samples infiltrated with SWNT-PM-CytKH9 with an approximately 10-fold increase in normalized G-band intensity relative to the samples infiltrated with SWNT-PM and SWNT-PM-KH9, suggesting that the dually functionalized SWNT was delivered most efficiently into the mitochondria (Fig. 3d and Supplementary Fig. 8). Taken together with the fluorescently-labeled SWNT results, these findings show that the SWNT NCs can be localized within plant mitochondria, with the Cytcox peptide conferring increased mitochondrial targeting.

**Luciferase expression and cytotoxicity in *A. thaliana* after infiltration with SWNT-PM-CytKH9/pDNA complexes.** To investigate the delivery efficiency of the pDNA within plants, we evaluated the expression of a pDNA encoding a *Renilla* luciferase reporter construct (*RLuc*) using the same vacuum/pressure infiltration conditions (0.08 MPa for 60 s) in the previous experiment within whole *A. thaliana* seedlings. To distinguish between expression from the mitochondria and the nucleus, two

previously cloned organelle-specific promoters were used (Fig. 4a)[13]. The Cauliflower mosaic virus (CaMV) 35S promoter (*pDONR-35S-RLuc*) was used to detect nuclear expression, and the mitochondrial-specific cytochrome *c* oxidase subunit II (Cox2) promoter (*pDONR-Cox2-RLuc*) was used to identify mitochondrial expression.

Based on DNA binding capacity of SWNT-PM-CytKH9, SWNT-PM-Peptide (400 µg) was complexed with 200 ng of pDNA for infiltration experiments (Fig. 2e, f). Both the 35S (19,000 ± 2700 RLU/mg protein) and Cox2 (34,000 ± 4500 RLU/mg protein) promoters showed the highest levels of luciferase activity for the samples infiltrated with SWNT-PM-CytKH9/pDNA (Fig. 4b) after overnight incubation of 18 h. This represents an almost 30-fold increase over previous results attained using the same pDNA construct and peptide-based delivery system, indicating a marked increase in delivery efficiency that is likely due to the penetrating abilities of SWNT[13]. In addition, the amount of expression observed in the mitochondria with the Cox2 promoter was approximately 1.5-fold higher than that in the nucleus with the 35S promoter, suggesting that the Cytcox peptide conferred increased selectivity to the NC for delivery into mitochondria. Furthermore, we observed no toxicity effects on the plants as quantified using the Evans blue viability assay (Fig. 4c) when incubated under the same conditions as the SWNT-PM-Peptide/pDNA infiltration, suggesting that the SWNT NCs were not toxic to the plants.

SWNT-PM-CytKH9/pDNA displayed approximately 1.5-fold higher expression values than those observed for SWNT-PM-Cytcox/pDNA (Fig. 4b). Based on the gel shift electrophoresis mobility assays (Fig. 2e, f), it is possible that this increase in efficiency could be due to higher amounts of DNA bound to the complex. However, no significant expression was detected for SWNT-PM-KH9/pDNA despite its having the highest pDNA binding capability during the binding assays. A potential explanation is that KH9 complexes the pDNA too tightly and is not efficiently released when the complex enters the mitochondria, which has a pH of 8. This explanation is supported by the fact that this complex was the most stable in a pH range of 5–9, as evidenced by the release of free DNA (Supplementary Fig. 9). Plants infiltrated with SWNT-PM without any conjugated peptide did not show any appreciable gene expression and were not significantly different from the DNA-only control. Taken together with the previous results, these findings show that the maximal organelle-targeted expression efficiency requires peptides with both cargo binding and releasing capabilities.

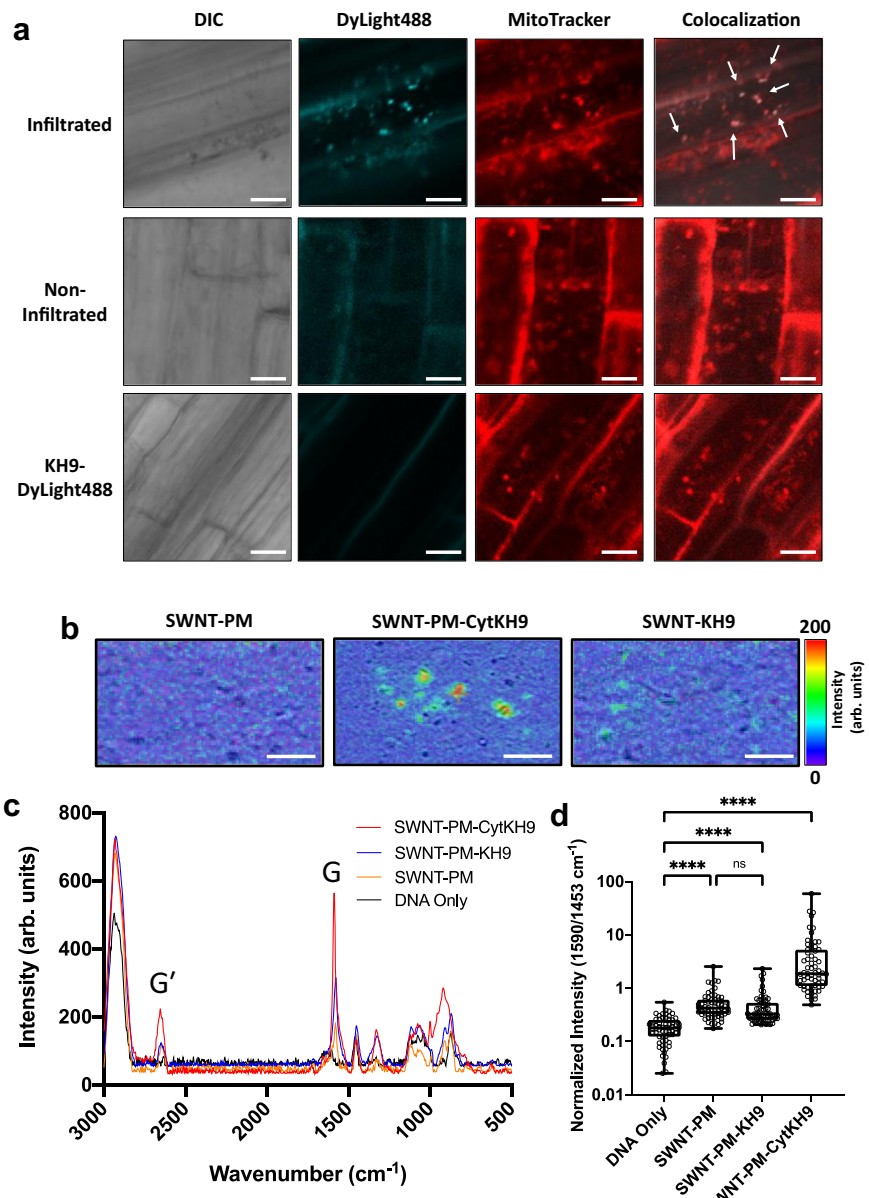

**Fig. 3 CLSM of fluorescently-labeled SWNT NCs and Raman microscopy of mitochondria in *A. thaliana* upon introduction of SWNT NCs.**
**a** Fluorescently-labeled (DyLight488) SWNT-PM-CytKH9 were detected within MitoTracker-labeled mitochondria and cytosol upon vacuum infiltration (upper panels) and on the surface of the root cells without vacuum infiltration (middle panels) when introduced into intact *A. thaliana* seedlings. White arrows indicate colocalization of the DyLight488-labeled SWNT-PM-CytKH9 and MitoTracker-labeled mitochondria. Minimal fluorescence was observed for the *A. thaliana* seedlings infiltrated with the labeled peptide only (bottom panels). Scale bars represent 10 μm. Representative CLSM images are shown from five imaged *A. thaliana* roots for each condition ($n = 5$) **b** Raman microscopy mapping of the 1590 cm$^{-1}$ G-band peak taken from isolated mitochondria isolated from *A. thaliana* 18 h after infiltration showed colocalization of SWNT signal with the isolated mitochondria in the samples infiltrated with SWNT-PM-CytKH9 and SWNT-PM-KH9. **c** Representative Raman spectra taken from mitochondria isolated from *A. thaliana* 18 h after infiltration show characteristic G (1590 cm$^{-1}$) and G' (2640 cm$^{-1}$) bands from the SWNTs. **d** Quantified normalized G-band peak (1590 cm$^{-1}$) intensities for isolated mitochondrial samples shown in (**c**). Relative values from 64 individual spectra ($n = 64$) are shown for each individual sample and the respective box and whisker plots depicting the maxima, minima, median and the first (25%) and third (75%) quartiles. Statistical significance between individual samples was determined by one-way ANOVA. *P*-values are <0.0001 for comparisons between DNA only and all SWNT-treated samples. ns – not statistically significant, ****$P$ < 0.0001. Source data are provided as a Source Data file.

**GFP expression within mitochondria of *A. thaliana* using SWNT-PM-Peptide/pDNA complexes.** To further confirm the introduction of pDNA within the mitochondria, we performed confocal laser scanning microscopy (CLSM) and western blotting analysis on isolated proteins from *A. thaliana* infiltrated with SWNT-PM-Peptide/pDNA complexes (Fig. 5 and higher magnification in Supplementary Fig. 10). Previously-constructed pDNA containing GFP reporter genes (*pDONR-35S-GFP* for

nuclear expression; *pDONR-Cox2-GFP* for mitochondrial expression) was used as reporting constructs for GFP expression analysis[12,13] (Fig. 5a).

GFP expression could be detected in the roots of samples 18 h after infiltration and showed efficient expression within the cytosol and mitochondria for both 35S and Cox2 promoters, respectively (Fig. 5b). No expression was observed for the samples infiltrated with pDNA exclusively. Colocalization analysis of GFP

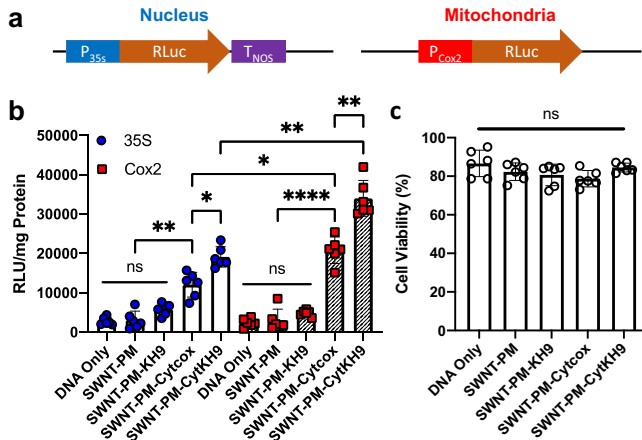

**Fig. 4 Transfection efficiency of the luciferase reporter construct and cytotoxicity of SWNT NCs. a** Reporter construct design of transient luciferase expression for the nucleus and mitochondria. **b** Transfection efficiency of the SWNT-PM-Peptide and pDNA was evaluated 18 h after infiltration using a Renilla luciferase reporter construct under the control of mitochondria-specific (Cox2) and nuclear (35S) promoters using whole seedling lysate soluble fractions. Data from 6 biologically independent samples ($n = 6$) are shown as the mean ± standard deviation. Statistical significance between individual samples was determined by one-way ANOVA with Brown-Forsythe and Welch test for multiple comparisons. *P*-values for the 35S promoter are 0.0087 between SWNT-PM and SWNT-PM-Cytcox, 0.0489 for SWNT-PM-Cytcox and SWNT-PM-CytKH9. For the Cox2 promoter, *P*-values are <0.0001 between SWNT-PM-SWNT-PM-Cytcox, and 0.0086 for SWNT-PM-Cytcox and SWNT-PM-CytKH9. For comparisons between 35S and Cox2 promoters, the *P*-values are 0.0265 for SWNT-PM-Cytcox, and 0.0033 for SWNT-PM-CytKH9. ns – not statistically significant, *$P < 0.05$, **$P < 0.01$, ***$P < 0.001$, ****$P < 0.0001$. **c** Evans blue assays for intact *A. thaliana* seedlings 18 h post infiltration with each respective NC, normalized to the absorbance of a boiled sample. *A. thaliana* infiltrated with DNA alone was used as a control. Data points from six biological replicates ($n = 6$) are represented as the mean ± standard deviation. Statistical significance between individual samples was determined by one-way ANOVA with Kruskal-Wallis test for multiple comparisons. ns – not statistically significant. Source data are provided as a Source Data file.

infiltrated with SWNT-PM-Cytcox/pDNA and SWNT-PM-CytKH9/pDNA, with SWNT-PM-CytKH9/pDNA having the highest GFP expression in both the nucleus and mitochondria. The addition of the KH9 peptide to SWNT-PM-Cytcox approximately doubled GFP expression for both promoter constructs and generally reflected the results seen in the luciferase assays (Fig. 5c, d). These results, along with the DNA binding and uptake studies, suggest that both Cytcox and KH9 were required to achieve optimal uptake and expression and that the peptides likely function in a synergistic manner. Cytcox significantly increased the mitochondrial targeting abilities of the SWNT NCs but had overall poor DNA binding capabilities (Fig. 2f).

By combining Cytcox with KH9 to increase the overall surface charge for DNA binding, we were able to optimize the SWNT NCs for directed uptake of the maximum amount of DNA within the mitochondria. This was further confirmed by the fact that although SWNT NCs were detected within the mitochondria using fluorescently-labeled SWNT NCs and Raman microscopy, the highest uptake was observed for SWNT-PM-CytKH9. We surmise that this is based on increased binding of pDNA as well as greater amounts of the complex entering the mitochondria, as suggested by the fluorescently-labeled SWNT-PM-CytKH9 and Raman experiments. We note that the SWNT NCs likely had a degree of nonspecificity and could localize within the nucleus in addition to the mitochondria, as evidenced by the nuclear expression with the 35S promoter, and the observed organelle-specific expression was from the promoters used in the respective constructs. Previous research into the subcellular localization of pristine and surface-modified CNTs has shown diverse localization, including cytosol, mitochondria, nuclei, or lysosomes, in mammalian cells, suggesting the general nonspecificity of CNT localization within mammalian cells[30,31]. Mitochondria-targeted peptides or PEG-modified CNTs used in mammalian cells previously showed uptake within the cytosol and lysosomes in addition to the mitochondria. The nonspecific localization of CNTs may have played a role in our observation of pDNA expression in both the cytosol and mitochondria despite the use of mitochondria-targeting Cytcox peptide[31,32]. Nonetheless, organelle-specific expression with SWNT NCs was achieved through the use of nucleus (35S)- and mitochondrial (Cox2)-specific promoters.

**SWNT-PM-CytKH9-mediated exogenous gene integration of folic acid metabolic enzyme into the mitochondrial genome showed enhanced root growth.** We then assessed the ability of the SWNT NCs to deliver pDNA competent for genomic integration into mitochondria under similar conditions using the pAtMTTF1 integrating construct with *SUL1*, which encodes dihydropteroate synthase type-2 from *Pseudomonas aeruginosa* and was previously used in *N. tabacum* and *C. reinhardtii* as a selection marker for mitochondrial transformation[33], and GFP reporter genes with mitochondrial *Cox2* promoters (Fig. 6a). SWNT-PM-CytKH9 complexed similarly with the pAtMTTF1 pDNA (10 kb) compared to the previously used *pDONR* pDNA (5 kb), suggesting that the SWNT NCs are able to complex with larger plasmids (Supplementary Fig. 14). SWNT-PM-CytKH9/ pDNA was prepared using 250 ng of pAtMTTF1 and 500 μg of SWNT and introduced into the plants as in previous transient luciferase and GFP expression experiments (Fig. 6).

PCR genotyping of tissue samples 7 days after infiltration showed that the exogenous sequence was successfully integrated into the mitochondrial genome (Fig. 6b). DNA integration events were observed in all samples treated with SWNT-PM-CytKH9/ pAtMTTF1 complexes, as indicated by positive genotyping PCR products for both the left and right homology arms (Fig. 6b).

expression and mitochondria stained with MitoTracker dye also showed no mitochondrial GFP expression for the sample of pDNA containing the 35S promoter or GFP expression within the mitochondria for samples infiltrated with pDNA containing the Cox2 promoter (Fig. 5b). These results indicated that the promoters within the respective pDNAs were sufficient to direct organelle-specific expression.

Cytosolic and mitochondrial protein fractions were then isolated from *A. thaliana* seedlings for expression quantification by western blotting (Fig. 5c–f). Mitochondria were confirmed to be isolated from *A. thaliana* by probing for the inner membrane protein cytochrome C and visualizing with CLSM after MitoTracker staining (Supplementary Fig. 11). Minimal contamination was observed in the cytosolic fraction, as neither mitochondrial protein was detected in appreciable amounts. GFP protein expression was detected in both the cytosolic and mitochondrial isolates as a single prominent band at 27 kDa when probed using anti-GFP antibody (Fig. 5c, d and Supplementary Figs. 12, 13). Quantitative analysis of the band intensities was performed by normalizing protein amounts to actin and cytochrome *c* abundance for cytosolic and mitochondrial proteins, respectively. The results agreed well with the luciferase assays, and GFP expression was detected only in the samples

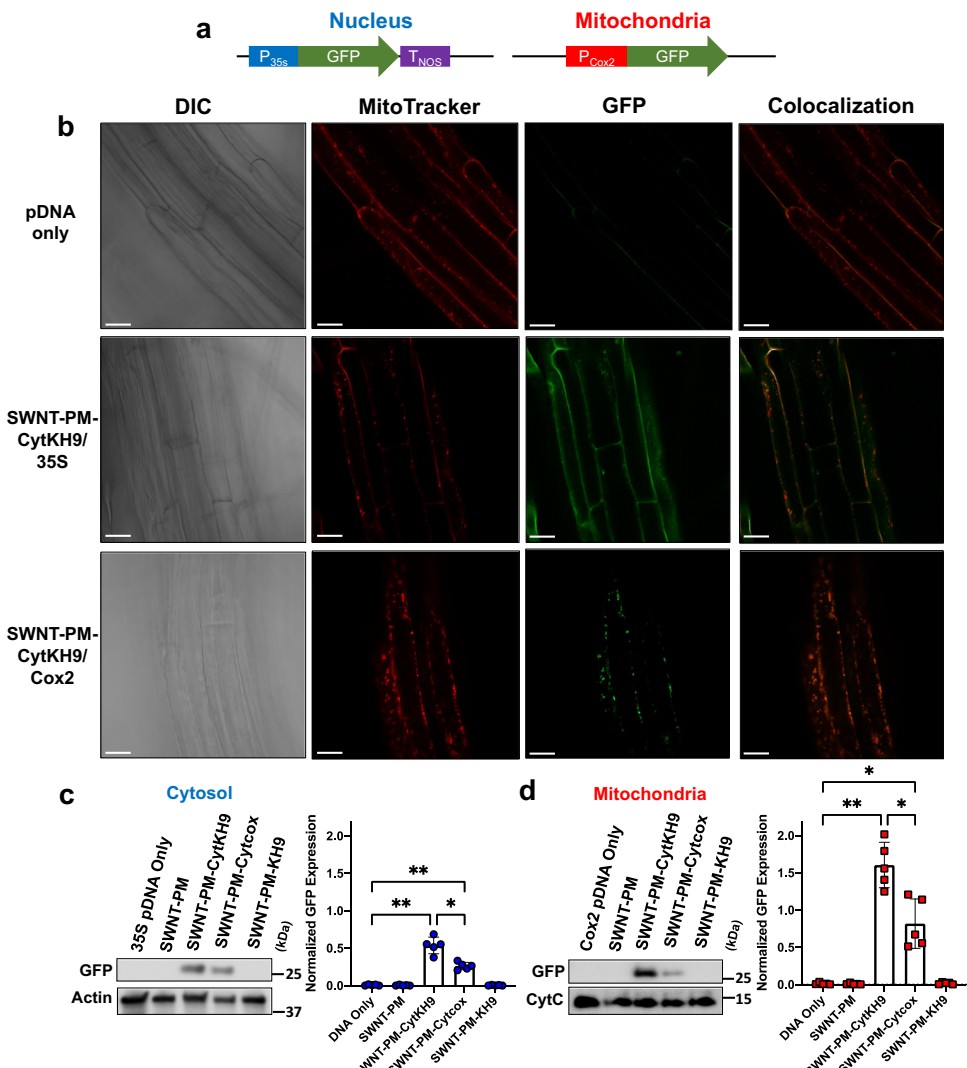

**Fig. 5 GFP Expression within *A. thaliana* upon infiltration with SWNT-PM-CytKH9/pDNA complexes. a** Reporter construct design of transient GFP expression for the nucleus and mitochondria. **b** Representative confocal laser scanning microscopy images of *A. thaliana* root cells from a minimum of five seedlings for each condition ($n = 5$) 18 h post infiltration with SWNT-PM-CytKH9/pDNA complexes containing *pDONR-35S-GFP* or *pDONR-Cox2-GFP* reporter constructs for nuclear or mitochondrial expression, respectively. Mitochondria were stained with MitoTracker Red (CMXRos), and colocalization analysis was performed on GFP expression and MitoTracker signals. Scale bars represent 20 µm. **c, d** Representative western blots and their quantification from cytosolic (**c**) and mitochondrial (**d**) fractions isolated from approximately 60 seedlings 18 h post infiltration with SWNT NCs probed for GFP expression. Actin and cytochrome *c* were used as loading controls for cytosolic and mitochondrial proteins, respectively. Data points from five biological replicates ($n = 5$) are represented as the mean ± standard deviation. Statistical significance was determined by Brown-Forsythe and Welch one-way ANOVA test. For the cytosolic samples, *P*-values are 0.0027, and 0.0020 between DNA only and SWNT-PM-CytKH9, SWNT-PM-Cytcox, respectively, and 0.0253 between SWNT-PM-CytKH9 and SWNT-PM-Cytcox. For the mitochondrial samples, *P*-values are 0.0019, and 0.0333 between DNA only and SWNT-PM-CytKH9, and SWNT-PM-Cytcox, respectively, and 0.0359 between SWNT-PM-CytKH9 and SWNT-PM-Cytcox. ns – not statistically significant, *$P < 0.05$, **$P < 0.01$. Source data are provided as a Source Data file.

These PCR products span the junctions between the mitochondrial genome and the homologous sequence from the donor plasmid (pAtMTTF1), and thus would only be amplified in the case of successful recombination. (Fig. 6b). In contrast, positive PCR products were detected faintly in only one sample out of six treated with pAtMTTF1 alone. Sequencing of the PCR products from the 5′ and 3′ arms of the construct confirmed integration at the expected positions within the mitochondrial genome (Supplementary Fig. 15 and Supplementary Data 1, 2).

The expression of the integrated reporter construct GFP was then assessed using CLSM. In the roots of *A. thaliana* seedlings treated with SWNT-PM-CytKH9/pAtMTTF1 complexes, colocalization of GFP signals with MitoTracker-stained mitochondria

was detected 1 day after transformation (Fig. 6c), similar to the results in seedlings transformed with *pDONR-Cox2-GFP* (Fig. 5b). GFP expression decreased slightly at 3 days after transformation and further at 7 days after transformation, but detectable GFP colocalization with the stained mitochondria was still visible (Fig. 6c). However, no mitochondrial GFP expression was detected in seedlings transformed with pAtMTTF1 pDNA only. These results indicate that SWNT-PM-CytKH9 can deliver DNA constructs for efficient recombination into the mitochondrial genome for stable integration and expression of transgenes.

*A. thaliana* seedlings treated with the SWNT-PM-CytKH9/ pAtMTTF1 complex also showed greater root growth (Fig. 6d and Supplementary Figs. 16–19) than seedlings treated with only

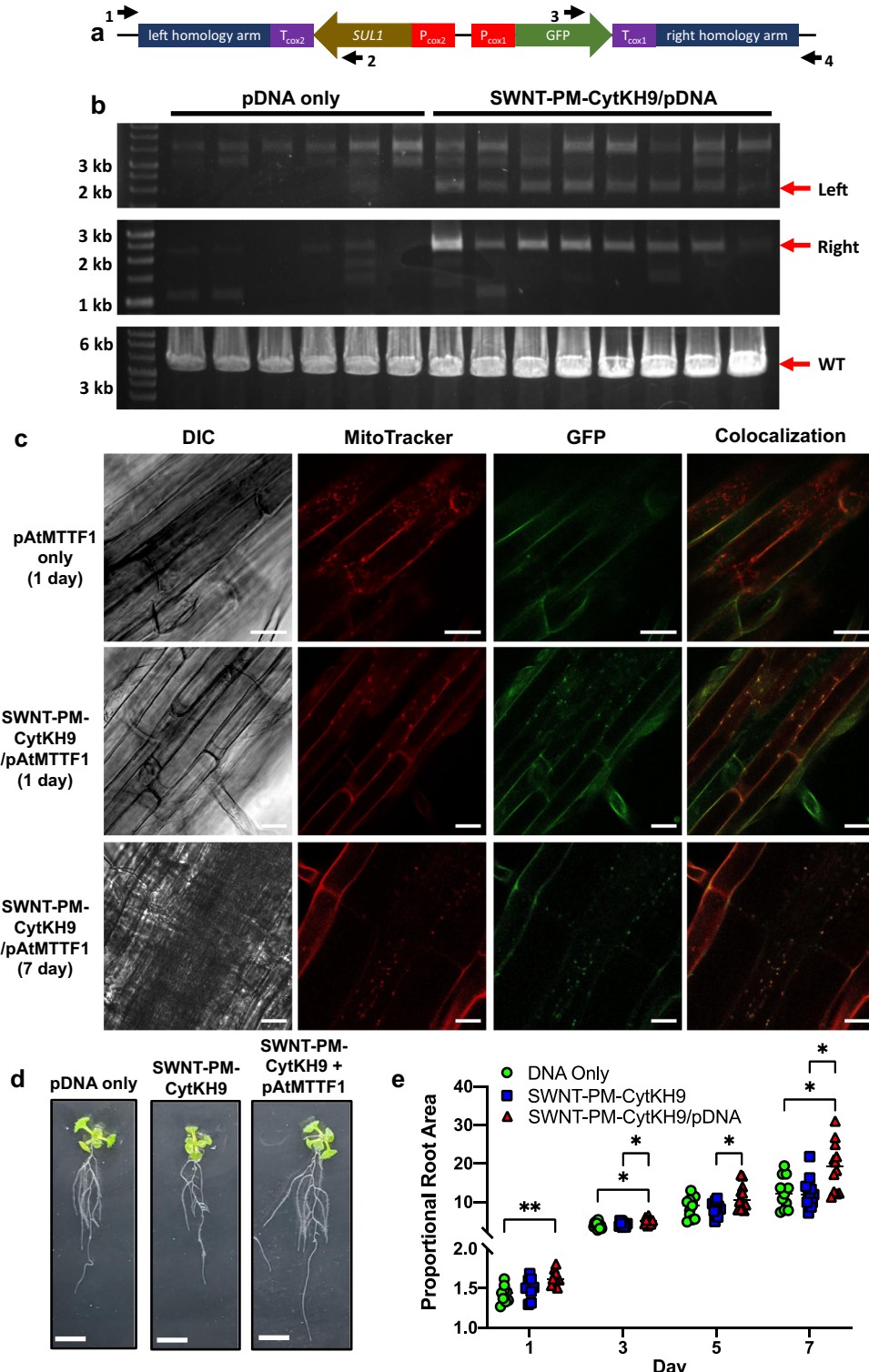

pAtMTTF1 or SWNT-PM-CytKH9, as quantified by the relative root growth in area over the course of 1 week after transformation (Fig. 6d, e). No significant difference was observed between the pDNA-only or SWNT-PM-CytKH9/pDNA complex treated seedlings when a negative control construct with the *SUL1* and its promoter and terminator removed (Supplementary Fig. 20).

Chlorophyll fluorescence and mitochondrial respiration measurements did not show any significant differences upon SWNT NCs infiltration, suggesting that the overall increase in root area was not due to overall metabolic changes and that the plants were

not stressed by the SWNT NCs introduction (Supplementary Fig. 21a, b). We then quantified intracellular folate concentrations, which showed significantly increased folate levels relative to the pDNA only and SWNT-PM-CytKH9/pDNA (-*SUL1*) infiltrated samples at days 1 and 3 post infiltration (Supplementary Fig. 21c). We hypothesize that the expression of *SUL1* from the transformed construct led to the observed increased folate levels, thus increasing the growth rate observed within the roots. In plants, dihydropteroate synthase functions in the mitochondrial steps of the folic acid biosynthetic pathway, producing

**Fig. 6 Genotyping and phenotypic effects in *A. thaliana* upon infiltration with SWNT-PM-CytKH9/pAtMTTF1 pDNA complexes. a** Design of a plasmid DNA construct (pAtMTTF1) for integration into the mitochondrial genome of *A. thaliana*. *SUL1* encodes dihydropteroate synthase type-2, which was previously used as a selection marker for mitochondrial transformation[36]. **b** PCR analysis of the exogenous DNA inserted into the mitochondrial genome 7 days after infiltration by primer pairs targeting the left or right junctions between the inserted DNA and the mitochondrial genome. The arrows indicate the size of the expected PCR products. Amplification of the wild-type locus is shown in the bottom panel. Raw sequencing data are provided in Supplementary Data 1 and 2. 6 and 8 representative samples are shown from a minimum of 30 biological replicates ($n = 30$) for pDNA only and SWNT-PM-CytKH9/pDNA treated samples, respectively. **c** Representative confocal laser scanning microscopy images of *A. thaliana* root cells (from a minimum of five seedlings for each condition $n = 5$) infiltrated with pDNA containing the pAtMTTF1 reporter construct for mitochondrial genome integration. Mitochondria were stained with MitoTracker Red (CMXRos), and colocalization analysis was performed on GFP and MitoTracker signals 1 and 7 days post transformation. Scale bars represent 20 μm. **d** Representative images of seedlings upon infiltration with SWNT-PM-CytKH9/pAtMTTF1 that showed *SUL1*-mediated growth improvement after 3 days. Scale bars represent 1 cm. Other days can be found in Supplementary Figs. 16–19. **e** Quantification of root areas of *A. thaliana* seedlings at 1, 3, and 7 days after transformation with SWNT-PM-CytKH9/pAtMTTF1 complexes relative to the initial area. Each data point represents one biological seedling replicate and their respective means for each treatment condition. Statistical significance was determined by two-way ANOVA (Tukey's multiple comparisons) with matched measurements for each replicate. For Day 1 samples, *P*-values are 0.001 between DNA only and SWNT-PM-CytKH9/pDNA. For Day 3 samples, *P*-values are 0.0312 between DNA only and SWNT-PM-CytKH9/pDNA and 0.0258 between SWNT-PM-CytKH9 and SWNT-PM-CytKH9/pDNA. For Day 5 samples, *P*-values are 0.024 between SWNT-PM-CytKH9 and SWNT-PM-CytKH9/pDNA. For Day 7 samples, *P*-values are 0.0190 between DNA only and SWNT-PM-CytKH9/pDNA and 0.0168 between SWNT-PM-CytKH9 and SWNT-PM-CytKH9/pDNA. *$P < 0.05$, **$P < 0.01$. Source data are provided as a Source Data file.

dihydropteroate from 6-hydroxymethyldihydropterin pyrophosphate and *p*-aminobenzoic acid. Folate end products have been shown to be required for postembryonic root development in *A. thaliana*, and increased folate levels have been shown to affect root growth in ways such as increasing lateral root proliferation[34,35]. These results suggest that the SWNT-PM-based transformation system shows potential for the metabolic engineering of mitochondria by direct and stable transformation of the mitochondrial genome.

CNT-based delivery methods have been gaining increasing traction in plant biotechnology, and bound cargoes display increased stability in both mammalian and plant systems[17,36,37]. However, organelle-specific delivery methods within plants are still lacking due to uptake limitations and relatively low delivery efficiency. This is further compounded for mitochondria targeted transformation due to the instability of the plant mitochondrial genomes and the lack of strong selection markers that can be maintained during the development of the adult plant after zygote formation. In this study, we have successfully tackled these problems by the use of peptide conjugated SWNT NCs with sufficient delivery and expression efficiency for genome integration and expression with an observable root phenotype. Further development of such a selection system would enable follow-up studies of maternal inheritance leading to genomic engineering of mitochondrial agronomic traits.

By combining SWNT and a cross-linked polymer with the functional peptides Cytcox and KH9, we developed a flexible and highly efficient CNT-based delivery method for plant mitochondria. Due to the non-covalent nature of the coated polymer, the SWNT retains its physical and optical properties, including its strong inherent Raman signal and unique aspect ratio, which allows effective cell wall penetration. Compared with previously reported peptide-based carriers, the SWNT-polymer hybrids achieved nearly a 30-fold increase in specific transient DNA expression.

Although the SWNT NCs also localized within the nucleus of the plant cells and could potentially limit their applications where mitochondrial-specific localization of the SWNT NCs are required, we believe sufficient specificity can be achieved by the tailoring the cargo such as by using organelle-specific promoters or translocation sequences. Nonetheless, this method holds tremendous promise for high-efficiency delivery into plants and can be customized further for different types of cargo and targets through modification of the polymer layer and peptides. Furthermore, we have successfully demonstrated that this method can be used for stable integration into the mitochondrial genome, paving the way for plant engineering applications with potential crop improvement and agronomic applications.

## Methods

**Synthesis of SWNT-PM and SWNT-PM-Peptide**. *Endo*-form furan-protected maleimide was synthesized from maleimide (0.855 g, 8.8 mmol; Tokyo Chemical Industry Co., Ltd., Japan) and furan (1.1 mL, 15 mmol; Tokyo Chemical Industry Co., Ltd., Japan) in water (10 mL), stirred at room temperature for 5 days, and collected by vacuum filtration. DEAD (0.65 mL, 1.43 mmol; Sigma-Aldrich, Japan) was added dropwise to the resulting product (200 mg, 1.25 mmol) with anhydrous THF (5 mL; Wako Pure Chemical Industries, Ltd., Japan), PPh₃ (375 mg, 1.43 mmol; Tokyo Chemical Industry Co., Ltd., Japan), and PEGMA (300 μL; Sigma-Aldrich, Japan) in an ice bath while stirring. The reaction was stirred at room temperature for 24 h. The solvent was removed by vacuum evaporation, dissolved in diethyl ether and extracted twice with water. The aqueous phases were extracted with chloroform three times. The combined organic phases were dried with MgSO₄ (Tokyo Chemical Industry Co., Ltd., Japan), and the orange oil was purified by silica gel flash column chromatography. *Endo*-FpMMA was obtained as a yellow oil and used to prepare polymer-coated SWNTs.

SWNTs (5 mg; Carbon Nanotechnologies, Inc., USA) were dispersed in 0.2% w/w SDS solution (50 mL) through ultrasonication for 1 h and centrifuged at 60,000x *g* (Hitachi Himac) for 1 h. *Endo*-FpMMA, *N,N*-methylene bis(acrylamide) (10 mg; Tokyo Chemical Industry Co., Ltd., Japan), and PEGMA (Sigma-Aldrich, Japan) were added to the supernatant (5 mL), and the mixture was bubbled with N₂ gas for 15 min. TMEDA (4.4 μL; Tokyo Chemical Industry Co., Ltd., Japan) and ammonium peroxodisulfate (20% w/w; Sigma-Aldrich, Japan) solution were added under a N₂ atmosphere and allowed to react for 24 h. The resulting solution was filtered through cotton, and the filtrate was dialyzed using a 10 kDa membrane (Merck Millipore, USA) for 3 days. Deprotection of the maleimide groups for thiol functionalization was achieved by heating at 80 °C for 2 h.

Mitochondria-targeting (Cytcox) and DNA binding (KH9) peptides (0.25 mM for Cytcox and 0.1 mM for KH9) were synthesized by the Research Resources Center of RIKEN Brain Science Institute and used to functionalize the deprotected maleimide at 16 °C for 24 h. Unreacted peptides and released monomer groups were dialyzed using a 3.5 kDa Slide-a-lyzer (Merck Millipore, USA) for 24 h to yield the respective SWNT-PM-Peptide and were stored at 4 °C prior to infiltration within *A. thaliana*. DyLight488-conjugated KH9 peptides were used to synthesize fluorescently-labeled SWNT-PM-CytKH9.

**Physical characterization of SWNT NCs**. Atomic force microscopy was performed on the SWNT-PM nanoparticles using a Hitachi AFM 5300E (Hitachi High-Tech Science Cooperation, Japan). SWNT nanoparticles were spotted on a graphite substrate (5 μL), evaporated at room temperature for 1 h and rinsed with Milli-Q water. The dried samples were then used for AFM observation in air at 25 °C using a silicon cantilever (SI-DF3, Hitachi High-Tech Science Corporation, Japan) with a spring constant of 1.7 N m⁻¹ in tapping mode. AFM height profiles were analyzed and quantified by Gwyddion (Version 2.58).

The zeta potentials and DLS measurements of the SWNT-PM nanoparticles were characterized using a Zetasizer Nano-NZ (Malvern Instrument, UK) and Zetasizer software ver. 7.12 using a sample volume of 700 μL for zeta potential and 100 μL for DLS measurements in a folded capillary Zeta cell (DTS1070, Malvern Panalytical, UK)[38]. The ζ potential and ζ deviation of samples were measured three

times by the same Zetasizer Nano-NZ, and the average data were obtained using Zetasizer software ver. 7.12 (Malvern Instruments Ltd.).

DNA binding of the SWNT NCs was evaluated by gel shift electrophoresis. 20 μL of SWNT-PM NC solution was incubated with 50 ng of pDNA (pDONR-35S-GFP) for 30 min, mixed with loading buffer before analysis on a 1% agarose gel in TAE buffer, and stained with ethidium bromide. Quantification of the plasmid band was performed using ImageJ (NIH).

**HPLC and MALDI-TOF/MS of SWNT NCs**. Peptides were cleaved from the SWNT polymer matrix using alkaline hydrolysis (1 M NaOH) at 80 °C for 1 h. The reaction was neutralized with 1 equivalent of HCl, and 100 μL was injected for analysis via reversed-phase high-performance liquid chromatography. The chromatograph was equipped with an autosampler (AS-2055, JASCO, Tokyo, Japan), gradient pump (PU2089, JASCO), column oven (CO-4060, JASCO), and C18 column (YMC-Triart C18, particle size 5 μm, 150 × 4.6 mm i.d., YMC, Kyoto, Japan). The samples were eluted using a gradient consisting of Milli-Q water (eluent A), acetonitrile (eluent B), and 1% (v/v) TFA (eluent C). The composition of the gradient varied linearly from 85% eluent A, 5% eluent B, and 10% eluent C to 55% eluent A, 35% eluent B, and 10% eluent C over a period of 30 min at a flow rate of 1 mL/min and a column temperature of 25 °C. The elution of the peptides was monitored by UV absorbance at 220 and 260 nm and analyzed using chromatography software (ChromNAV, JASCO, Tokyo, Japan). The fractions corresponding to the respective peptides were collected and lyophilized for MALDI analysis.

MALDI-TOF MS spectra were recorded on a MALDI-TOF spectrometer (Autoflex speed LRF; Bruker Daltonics, Billerica, MA, USA) operating in positive ion reflection mode at a 15 kV accelerating voltage. Samples were prepared by mixing the cleaved PM or peptides with α-cyani-4-hydroxycin-namic acid (10 mg/mL) and trifluoroacetic acid (TFA) (0.1%) in a 1:1 ratio (5 μL total). The samples (2 μL) were deposited on an MTP 384 ground steel BC target plate and dried *in vacuo* for 1 h before measurement. Raw mass spectral data are provided as Supplementary Data 3–5.

**MD simulations of the SWNT-PM**. A CNT structure 20 nm long was generated using the Python (Version 2.7) script BuildCstruct and GAFF parameters assigned to the running acpype[39,40]. The polymer was constructed from monomer units, requiring the construction and parametrization of a central unit with two available bonds and a head and tail unit with one available bond to connect with a central unit. The monomer units were constructed using Gaussview, and geometry optimization was performed with quantum mechanics (QM) at the B3LYP/6-31 G* level of theory using Gaussian16[41]. The restraint electrostatic potential (RESP) charges were calculated with the Hartree-Fock/6-31 G* basis set with Gaussian16. For the geometry and energy calculations, the units were capped with methyl groups. The Antechamber module in AMBER18[42] was used to RESP fit the calculated potentials and strip the charges of the methyl units used for capping the monomers to generate amber files.

Systems were constructed with the Antechamber module of AMBER, and all MD simulations were performed using AMBER 18[42]. The atoms were described with GAFF2, and the system was solvated in a box of TIP3P water molecules at least at 10 Å of any atom[43]. A time step of 2 fs was used with the SHAKE algorithm[44], and the particle-mesh Ewald (PME) method[45] was used to calculate long-range interactions. The water molecules and then the whole system was energy-minimized for 20,000 steps using the conjugate gradient minimization algorithm; then, the system was slowly heated to 300 K over 600 ps with the solute atoms fixed with a 2 kcal mol-1 restraint. Then, the system was equilibrated in the NVT ensemble over 600 ps to ensure the appropriate density of the water box. The simulations were extended for 150 ns until the system was considered equilibrated according to the root mean square deviation (RMSD) (Supplementary Fig. 7c). Visualization of the MD results was performed using PyMol 1.8.5.

**Plasmid DNA preparation**. Plasmids containing the GFP and luciferase reporter constructs were previously prepared and were shown to express exclusively within the mitochondria of *A. thaliana* (Plasmid maps are included in Supplementary Figs. 22–24)[12,13,46]. The *pDONR-Cox2p-RLuc* and GFP constructs contain the promoter from *S. cerevisiae* COX2 and the reporter construct, without a terminator sequence. Previous studies have demonstrated that deletion of the untranslated terminator sequence did not affect expression of pDNA containing reporter constructs introduced within plant mitochondria[47,48]. The pAtMTTF1 pDNA constructs were designed to express the *SUL1* resistance gene and a *GFP* gene under the control of the *S. cerevisiae* cox2 and *A. thaliana* cox1 promoters, respectively (Fig. 6a). The pAtMTTF1 no-sul (-*SUL1*) pDNA construct is similar to pAtMTTF1 but with the *SUL1* coding sequence and its promoter and terminator deleted. Each construct has gene sequences flanked by two homologous regions from the *A. thaliana* mitochondrial genome, each approximately 1.6 kbp in length. The homologous regions in pAtMTTF1 correspond to positions 167,263 to 168,788 and 165,539 to 167,260 in the *A. thaliana* mitochondrial genome. This construct is expected to be integrated into the mitochondrial genome via homologous recombination, which occurs readily within plant mitochondria[49,50].

**Plant growth conditions and delivery of SWNT-PM-Peptide/pDNA**. Seeds of *Arabidopsis thaliana* Col-0 were germinated on 0.5x Murashige and Skoog (MS) (Sigma-Aldrich; USA) medium plates under 16-/8-h light/dark periods at 22 °C for *A. thaliana*. SWNT-PM-Peptide/pDNA complexes were prepared by mixing 500 μg (5 mg/mL) of SWNT-PM-Peptide solution and 250 ng of pDNA at 25 °C for 30 min. Seven-day-old *A. thaliana* seedlings were used to assess the delivery of pDNA and its expression and integration in *A. thaliana*. Vacuum/pressure infiltration was performed by incubation of whole seedlings (10 μL solution per seedling) in the respective solution and subjected to 0.08 MPa vacuum followed by pressure for 60 s each. The seedlings were allowed to recover in the solution for 1 h at room temperature before being plated on 0.5x MS medium plates. For root growth measurements, the plates were placed at an orientation of approximately 75°. Seedlings were allowed to grow under 16-/8 h light/dark periods at 22 °C for 1–7 days.

Uptake of SWNT-PM-Peptide and expression of pDNA containing GFP and luciferase reporter constructs were assessed after 18 h. For seedling growth and genotyping experiments, seedling growth was assessed at 1- to 2-day intervals, and samples were collected at various timepoints for PCR genotyping and confocal laser scanning microscopy (CLSM) analysis, respectively.

***Renilla* luciferase and Evans blue viability assays**. The reporter construct *RLuc* gene expression was quantitatively evaluated using the *Renilla* luciferase assay (Promega, Madison, WI) and was performed ($n = 6$) according to the manufacturer's protocol. Infiltrated seedlings were incubated for 18 h and homogenized before lysis with *Renilla* Luciferase Assay Lysis Buffer (Promega). The lysate was centrifuged for 3 min at 13,000 xg, and the supernatant was mixed with *Renilla* Luciferase Assay Substrate and Renilla Luciferase Assay Buffer (Promega) in a 1:1 ratio. Gene expression was evaluated based on the intensity of photoluminescence (relative light units) using a luminometer (GloMax 20/20, Promega). The amount of protein in the supernatant was determined using a Bradford protein assay (Pierce Biotechnology, Rockford, IL), and the ratio of relative light units/weight of protein (RLU/mg) was quantified for each sample. Background corrections were performed by subtracting the average RLU/mg value of a sample of untransfected seedlings.

**CLSM and western blotting**. GFP expression and localization of the labeled SWNT-PM-CytKH9 within *A. thaliana* after infiltration with the respective SWNT NCs was evaluated qualitatively using CLSM[12,13]. Seedlings vacuum infiltrated with the respective SWNT NCs were washed with dH$_2$O prior to staining for CLSM analysis (18 h incubation post infiltration for expression and 3 h for labeled SWNT). Mitochondria were stained using 100 nM MitoTracker Red CMXRos (Thermo Fisher, USA) for 1 h at room temperature and washed with dH$_2$O three times to remove excess dye before imaging[51]. Intact seedlings were placed on a microscope slide and suspended in dH$_2$O for CLSM measurements. Labeled mitochondria within the roots of *A. thaliana* were imaged using an excitation and emission (Ex/Em) wavelengths of 555/580–610 nm (for CMXRos). The intracellular localization of the GFP expression and DyLight488 were imaged at 488/500–540 nm and 488/500–540 nm, respectively.

For protein analysis, the mitochondrial and cytosolic fractions were separated from isolated roots of *A. thaliana* according to a modified protocol using a mitochondria isolation kit with the changes outlined as follows (Thermo Fisher Scientific, USA)[52]. Roots were isolated and homogenized from approximately 60 seedlings that had been infiltrated with the corresponding SWNT complexes that had been incubated overnight before following the isolation kit procedures. Proteins from the corresponding mitochondrial and cytosolic fractions were concentrated using a centrifuge concentrator (10 kDa cutoff; Merck Millipore, USA), and the resulting concentrates were lysed at 100 °C with SDS-PAGE loading buffer at a 1:3 v/v ratio (Bio-Rad Laboratories, USA) for 30 min before separation on a 4–15% gradient gel for SDS-PAGE (Bio-Rad Laboratories, USA). The gel was then transferred to a PVDF membrane (Bio-Rad Laboratories, USA) using a Mini Trans-Blot SD Semi-Dry Electrophoretic Transfer Cell (Bio-Rad Laboratories, USA) and blotted in 5% skim milk in PBST buffer overnight.

GFP was then detected using a rabbit polyclonal anti-GFP primary antibody NB600-308 (1:2500; Novus Biologics, USA) and goat anti-rabbit IgG conjugated with horseradish peroxidase (HRP) (1:20000; Abcam, UK). The actin loading control for the cytosolic fraction was probed with a mouse monoclonal anti-actin antibody A0480 (1:2500; Sigma-Aldrich, USA) and goat anti-mouse IgG conjugated with HRP (1:20000; Abcam, UK). The cytochrome *c* loading control was probed using a rabbit polyclonal anti-cytochrome *c* antibody AS08 343 A (1:4000; Agrisera, Sweden) and the previously used goat anti-rabbit IgG HRP-conjugated antibody. Quantification of each of the respective bands was performed using ImageJ (NIH, USA), and analysis was performed using GraphPad Prism Version 8.0 (GraphPad, USA).

**PCR genotyping of transformed plant tissues**. Total DNA was extracted from seedlings with the DNeasy Plant Mini Kit (QIAGEN) according to manufacturer's instructions. Primers were designed to amplify the wild-type mitochondrial genomic locus containing the integration sites for pAtMTTF1, outside of the homology arms. Primers were also designed (Supplementary Table 1) for

detecting the presence of the recombined construct, such that one primer binds outside of the homology arm and the other primer binds within the coding sequence of a gene in the construct, spanning either the left or right junction. Primers were also designed for amplifying the integrated construct, spanning either the left or right homology arm of pAtMTTF1. PCR was performed using PrimeSTAR GXL polymerase (Takara), with an annealing temperature of 64 °C and 35 amplification cycles for the left and right arm reactions, and 60 °C and 30 cycles for the wild-type reaction. DNA sequencing of the corresponding extracted bands was performed using the respective primers in Fig. 6a and included as Supplementary Data 1–2.

**Quantification of root growth and folic acid concentrations**. Seedlings were photographed at several points during incubation on MS medium plates. The photos were manually processed in ImageJ using the default threshold algorithm to detect roots. Root area was calculated as a fraction of the total area of a selection enclosing the root. The root area of each seedling at each timepoint was normalized to the initial root area on Day 0. Folic acid quantification was performed by extraction from six seedlings at Day 1, 3, and 7. Extraction was performed by grinding 12 seedlings in liquid nitrogen using a mortar and pestle and incubation in PBS buffer for 20 min. Folic acid quantification was performed according to kit instructions using the folic acid ELISA quantification kit (Cell Biolabs Inc., USA) and normalized to the respective sample protein concentrations.

**Chlorophyll fluorescence and mitochondrial respiration measurements**. Chlorophyll fluorescence was measured using Closed FluorCam FC 800-C (Photon Systems Instruments, Czech Republic) according to manufacturer's instructions. Briefly, plates of *A. thaliana* seedlings under each treatment condition were conditioned in the dark for 30 min before measurements. Standard Kautsky effect measurements were used to estimate $F_v/F_m$ values with a 2 s pulse time. Measurements were repeated after conditioning in the dark for 30 additional minutes and averaged. Analysis areas were selected by the FluorCam software (Version 7) with a minimum of 15 pixels and total values were weight-averaged by area for each plate.

Mitochondrial respiration was measured by ATP production rates in isolated mitochondria following the methods described in the Western blot analysis using the mitochondria from 20 seedlings. ATP quantification with the Cell-Titer Glo Luminescent Assay Kit was performed according to manufacturer's instructions. Gentamycin was used as a negative control to confirm cellular respiration and the rates of ATP generation were normalized by the respective sample protein concentrations.

**Raman microscopy and FE-SEM sample preparation and imaging**. FE-SEM samples were prepared by evaporation of the respective SWNT NC solution on a silicon wafer support at atmospheric pressure. For observation, the acceleration voltage and working distance were set to 2–6.0 kV and 2.0 mm, respectively. Images were captured with SmartSEM (Version 6.01) (Carl Zeiss, Germany).

Raman microscopy analyses of isolated mitochondria samples and the SWNT NCs were performed using a JASCO NRS-4500 Raman microscope (JASCO, Japan). Nanoparticle characterization was performed using aqueous samples at 100x on a cover glass with a green laser at 532 nm with an integration time of 6 s per spectra that were collected over an area of 50 × 40 μm. Isolated mitochondrial samples were prepared similarly for the western blot analysis using freshly prepared and isolated mitochondria that were imaged as aqueous samples on a glass slide immediately after isolation using the same conditions. Raman mapping studies were performed using a Raman Touch confocal Raman microscope (Nanophoton Corp., Japan) using a green laser at 532 nm collecting in line mapping mode.

**Statistical analysis**. GraphPad Prism 8.0 (GraphPad, USA) was used for statistical analysis. Statistical tests used to compare samples are listed in their respective figure legends. Differences between two means were considered statistically significant at $P < 0.05$ and are indicated with asterisks (*) in the plots. Exact $P$-values are also listed within the respective figure captions. Data in experiments are expressed as the means ± standard deviation unless otherwise noted, and sample sizes are stated in each respective figure.

**Reporting summary**. Further information on research design is available in the Nature Research Reporting Summary linked to this article.

## Data availability

The reporting summary for this article is included in the Supplementary Information. All data supporting the findings of this study are available within the article, the Supplementary information or the Source Data file. Mass spectrometry data and PCR sequencing data are included as Supplementary Data. Source data are provided as with this paper.

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

## Acknowledgements

We acknowledge the Support Unit for Bio-Material Analysis, RIKEN Center for Brain Science Research Resources Division, for performing the peptide syntheses and DNA sequencing and the Gene Discovery Research Group, RIKEN Center for Sustainable Resource Science, for the chlorophyll fluorescence experiments. We also acknowledge the RIKEN Information System Division for providing access to the Hokusai supercomputer for the molecular dynamics calculations. K.N. acknowledges the support from Japan Science and Technology Agency ERATO (Grant Number JPMJER1602), COI-NEXT, and the Grant-in-Aid for Transformative Research Areas (B) (Grant No. JP20H05735). T.F. thanks the support from Nakatani Foundation for Advancement of Measuring Technologies in Biomedical Engineering.

## Author contributions

S.SY.L., Y.K., T.F., and K.N. conceived the study and designed the experiments. S.SY.L., Y.N., and K.T. synthesized the SWNT NCs. S.SY.L. and G.L. performed the character-ization and the delivery experiments in plants. S.SY.L. and A.T. performed the FE-SEM experiments. J.G.D. performed the molecular dynamic simulations. S.SY.L. and G.L. analyzed the data. S.SY.L, G.L, and K.N. wrote the paper with input from all the authors. All authors have approved the final version of the manuscript.

## Competing interests

The authors declare no competing interests.
