## [Peer Review File · Nature Communications]

Polymer-coated carbon nanotube hybrids with functional peptides for gene delivery into plant mitochondriaReviewers' Comments:

Reviewer #1:

Remarks to the Author:

The manuscript describes the transformation of mitochondria (with protein and nucleic acid) using a carbon nanotube approach. This is a useful and novel approach, and with some significance as plant mitochondria remain one of the challenges to transform. While overall I am convinced by the material presented to transform mitochondria, a significant increase in efficiency on previous reports I do have some clarifications.

- 1) While it is great to show mitochondria are transformed and that the DNA was integrated, is it inherited into the next generation – this would really show transformation compared to transient transformation. Moreover given the almost universal maternal inheritance of mitochondria, reciprocal crosses would be needed to carry out to show that it is a maternally inherited trait.
- 2) I was interested in the fact that the transformed plants grow better – this is an important finding, and while it cannot be fully explained in this manuscript, cannot be left without more explanation. I am presuming the plants transformed are not limited by nutrient etc, especially folate. So why are they growing better? Some basic measurements of mitochondrial activities (and photosynthetic parameters) are warranted.
- 3) A number of mutants in nuclear genes encoding mitochondrial proteins or mitochondrial located genes encoding proteins that result in male infertility have been reported in Arabidopsis. The complementation of at least one of these by expressing in mitochondria would show the utility of this system.

In summary as presented the manuscript reports a substantial increase in efficiency for plant mitochondrial transformation. However, as this is not the first time plant mitochondria have been transformed, some useful utilisations of this system are necessary to show it can be inherited and address some of the issues that the authors raised in the introduction – otherwise it would be a rather specialised system with limited uses.

Reviewer #2:

Remarks to the Author:

see attached

Results summary:

The work by Lau et al uses polymer coated carbon nanotubes modified with peptides to enable plasmid delivery into the mitochondria of plants. The team uses different reporters to validate their platform and are able to quantify protein expression with western blots and fluorescence microscopy. The concept and advance is very high in its significance for mitochondria transformation in plants, which is likely unprecedented. However, given its potential very high impact, there are several points that should be addressed by the authors prior to publication as noted below.

Main points:

- Colorimetric assays such as evans blue relies on absorbance and emission of dyes. It is known that SWNT have very broad absorption and emission spectra that overlap with those of dyes. qPCR of plant stress genes should be performed for molecular-level validation of biocompatibility for the SWNT-PM-Peptide/pDNA complexes.
- Authors claim that the Cytcox peptide confers specific mitochondrial delivery while the data suggest that SWNT location is unspecific. SWNT-PM-KH9 in SFig6 are shown to be localized to mitochondria while there is no targeting peptide in this material (so localization should not be specific). Authors show that SWNT-PM-CytKH9 are able to deliver pDNA to both mitochondria and nucleus for expression. From the data they present, organelle-specific expression is given by the promoter in the pDNA, not by the Cytcox peptide, I think the text should explicitly reflect that. Their data clearly show that the combination of Cytcox and KH9 is what gives the highest expression, making CytKH9 a **novel** and very versatile peptide for delivery, not restricted to mitochondria. Because authors already have plasmids with chloroplast promoters, I would suggest that they try to deliver them and see if they detect expression.
- Regarding the claim of HDR in mitochondria, the PCR should be better explained and the sequencing data should be provided in the SI. Moreover, if there is HDR, the bottom panel of figure 6b should show a larger band corresponding to the locus with the introduced genes. Also, to sustain the claim that increased folate results in root growth, I would suggest that they show root folate quantification. Alternatively, a good negative control would be to deliver pDNA with only the GFP reporter lacking the *sul1* gene.
- Based on the provided FE-SEM micrographs, there are no SWNTs visible. The red arrows are pointing to features we see prolifically in plant leaf tissue EM of inclusion bodies. SWNT and peptides are carbon based and are therefore highly unlikely to be observable inside plant cells except in the vacuole. I suggest removing the SEM micrographs as they do not help support the claim of internalized SWNT. To this end, the resolution of confocal Raman (unspecified) is unlikely to resolve SWNT present in the mitochondria against those present in the cell or on the surface of the mitochondria or more broadly in plant tissue. Also please clarify why the baseline Raman intensity for the different SWNT treatments varies so greatly.
- Figure 6a-b: It seems that given the design of the primers, a PCR of the delivered pDNA complex (not necessarily the mitochondrial genome) would give rise to amplification of the same region. It may be that the PCR amplicon has amplified to a greater extent with

the SWNT sample because SWNTs are able to better protect the pDNA from degradation. It might be necessary to sequence regions that begin past the left and right homology arms to determine whether authors are amplifying a region from the pDNA or a mitochondrial genomic region.

Minor points:

- Root expression is particularly exciting. However I am unclear how root expression assays were done: were the roots vacuum infiltrated? Were these roots from plants that were leaf infiltrated? Hydroponic assays? Line 237 simply says “after infiltration” but it is unclear what was infiltrated and when.
- Authors show that loading of the plasmid generates a closer to neutral SWNT-plasmid suspension. Authors should test whether a larger plasmid could be adsorbed to the SWNT (not suggesting all of the study be redone, just the gel shift assays and colloidal stability tests).
- Figure 4 figure nor caption mentions what the biological substrate is, please mention it is leaves (I am assuming?)
- How come 200 ng of pDNA was loaded on the SWNT-PM-peptide complex? Please include reasoning in line 203.
- Line 48: Carbon nanotubes can be made **from** graphite but are not made **of** graphite.
- Figure 2a-c: It would be helpful for visual comparison if you could standardize the size of the scale bars across all samples, and also if the scale bar were smaller in size (100 nm or smaller).
- Figure 2a-c: From the images it looks like the width of the SWNTs are quite large, quite a bit larger than unmodified SWNTs. Do you know what the average width is, especially across samples?
- Figure 2g: Please include zeta potentials of the different SWNT samples after DNA has been loaded, I imagine the zeta potential would be highly dependent on amount of DNA.
- Supplementary Fig. 2: From what I understand, SWNT-PM-CytKH9 is derived from SWNT-PM, so it would be logical that mass % remaining at 500C (that is, what we attribute to pure SWNT) would be lower for the SWNT-PM-CytKH9 sample than SWNT-PM. Why is this not so?
- Please number your Supplementary Figures in order of appearance (10 comes before 9 in the manuscript).
- Figure 5b: Please provide scale bars on your images. Also, it would be great to have some higher magnification images available.
- Throughout the entire manuscript, authors should state prior to every result how many days-post-infiltration (dpi) their results were resultant from.
- It is unclear from initial reading of the text and figure 1 whether the peptides are attached to the SWNT covalently or noncovalently. I suggest reworking the text and figure accordingly. I.e. line 74-27 refers to prior publications but this is critical

information that needs to be included in this manuscript (it isn't until line 327 that authors mention that the polymer is noncovalently coated on SWNT).

- In general, the authors suggest that a mitochondrial localization sequence tethered to these relatively large nanostructures enables trafficking into the mitochondria. Are the entire SWNT structures, which can apparently be on the same size magnitude as mitochondrion themselves be trafficked in their entirety into these organelles? Or are the SWNT constructs somehow shedding these peptides and pDNA cargo in the cell? It would be good for authors to speculate, in addition to providing information on the average length and polydispersity of the SWNT used.
- Line 204. It is surprising that a human derived COX2 mitochondrial promoter is active in plant mitochondria. Are there literature to support this inter-kingdom adaptability? Are there any possible sources of non-specific transcription from this promoter? Additionally, the COX2 promoter elements are quite complex in their organization. How was the exact sequence to use determined? Additionally, could the authors explain the use of tNOS in their mitochondrial plasmid construct? It is surprising that a terminator used in nucleus expression is effective in this scenario as well.
- Could authors cite literature for the statement that HDR occurs readily within plant mitochondria?
- Line 286. In the literature cited for this selective marker, it appears that the sul1 protein was expressed by nucleus expression and then subsequently targeted to the mitochondria using a localization sequence. It was not developed for mitochondrial transformation. Is the transgene used in this experiment the same? If so, could there be erroneous nuclear expression and subsequent transport to the mitochondria that are driving the observed increase in folate?
- Line 293. In the mt genotyping experiment, could the authors provide the primer sequences used as well as that of the donor plasmid? It is unclear if the authors used primers that would span the gaps of the HDR template and the mtgenome itself or if regions within the plasmid were used. If it is the latter case, could the amplified pcr product not originated from the delivered plasmid itself? Additionally, for plastid transformation, and HDR in mammalian cells, a linear donor template is preferred so it is surprising a circular construct is effective. Actual sequencing of an amplicon that spans the gaps of the HDR donor within the mtDNA would strongly supplement the claims made by the author here.

Reviewer #3:

Remarks to the Author:

In the present paper the authors designed single-walled carbon nanotubes (SWCNT) for gene delivery in mitochondria. In particular they reported a detailed chemico-physical characterization of SWCNTs functionalised with peptides and plasmids. They employed two peptides to confer mitochondria targeting properties to SWCNT and increase DNA binding on the nanotubes. Then they checked the pDNA binding properties of this material before infiltrating them in plantlet of *A. thaliana*. They also proved that SWCNT-CA-pDNA enable expression of reporter genes in *Arabidopsis* and drive genetic integration through the homologous recombination of *sul1* gene involved in the folate pathway. Nanomaterials for genetic engineering are highly appealing although a technological gap still limits their use in plant. The paper has the merit to provide a new strategy for SWCNT-mediated transformation in plant mitochondria. The chemical-physical characterization of the nanomaterials is solid while some flaws are in the description of the uptake mechanisms. The specificity of mitochondrial transformation is not ensured but the approach sounds interesting. The significance and the novelty of the manuscript is suitable for publication on *Nature communications*. However, before publishing the following points need to be revised:

Abstract

Line 26: which method do they refer to?

Introduction

Line 81 - Information about Cytcox are not exhaustive and the reference of its targeting properties in plant is not appropriate. Similarly, KH9 details and relative references are missing

Results

Lines 87-93 This sentence appears confused and should be rephrased

Line 124. Please include a map of this plasmid

Line 130. Looking at the graph in Fig. 2f (blue curve), the ratio SWNT-PM(KH9)/ pDNA (1 microgram/ 50 ng) seems not well estimated.

Lines 134-137 What about the zeta potential of SWCNT-PM after plasmid binding? It should be reported.

Lines 173-176. FE-SEM images of infiltrated plants must be integrated with micrographs of non infiltrated plants treated with nanotubes and not infiltrated. FE-SEM are not totally convincing and could not discriminate clearly SWCNT-PM from naturally occurring granules available in the organelles. Therefore, the author must prove with additional techniques the mitochondrial localization. For instance, TEM images reported in literature could be much more useful than SEM to prove mitochondria uptake of SWNTs as reported in:

Ma X., et al 2021 ACS Nano. 2012;6(12):10486-10496. doi:10.1021/nn302457v

Battigelli et al., *Nanoscale*, 2013, 5, 9110-9117

- Alternatively, fluorescently -labelled nanotubes could be employed for mitochondrial colocalization analyses by confocal microscopy.

- Moreover, the uptake analysis must be strengthened by quantitative data, for instance by reporting the number of SWCNT positive mitochondria vs the total observed in the micrographs.

- Finally, the uncertainty to recognise SWNT from other organelle granules (as admitted by the authors at lines 178 -180) suggests to clearly state in the text that the SWCNT localization in other organelles cannot be excluded. The expression of nuclear promoters supports this.

Conclusion

Apart the valorisation of the results, the conclusion must mention also the limit of nuclear expression that could be a problem for those applications that may require specific mitochondrial expression. The author should include these considerations for proper information.

Responses to the Reviewers' Comments

We thank the reviewers for their valuable comments to improve our manuscript and the opportunity to revise the manuscript. A summary of additional figures and changes to original figures precede our detailed reply to the reviewers' individual comments. Our replies are colored in red with direct changes to the main text highlighted in yellow. Relevant figures and supplementary information have also been reproduced.

Summary of additional figures and changes to existing figures

Fig. 2a: Scale bars have been standardized across the images and the magnification of the SWNT-PM image has been increased to match the other images.

Fig. 2e: The crop of the gel image for KH9 has been moved slightly and the entire uncropped gel image has been added to Supplementary Fig. 5.

Fig. 2g: Zeta potential for the respective SWNT NC in complex with pDNA has been added.

Fig. 3: FE-SEM images have been replaced with CLSM images of *A. thaliana* infiltrated with DyLight488 labelled SWNT-PM-CytKH9 to confirm the localization within mitochondria. Furthermore, additional experiments using Raman mapping of isolated mitochondria are now included to show the colocalization of the SWNT G-band peak with the mitochondria. The spectra in the Raman spectroscopy graph were originally shifted in the y-axis for clarity but have been reverted to the original positions.

Fig. 4a: The NOS terminator sequence has been removed from the diagram for the Cox2 driven RLuc reporter construct. Plasmid sequencing showed that the NOS terminator was absent.

Fig. 5a: The NOS terminator sequence has been removed from the diagram for the Cox2 driven GFP reporter construct. Plasmid sequencing showed that the NOS terminator was absent.

Fig. 5b: Appropriate scale bars have been added to each CLSM image panel. Higher magnification images are now included in Supplementary Fig. 10.

Fig. 5c-d: The source images for the Western blots are included in Supplementary Figs 11 and 12.

Fig. 6a: Primers 1 and 4 locations have been moved in the construct diagram to better illustrate they bind to the mitochondrial genome, outside the homology arm of the integrating construct, and amplification can only occur if the gene is successfully integrated into the genome. Original sequencing data have also been included (Supplementary Fig. 14 and Supplementary Data 1 and 2).

Supplementary Fig. 2: Thermogravimetric analysis (TGA) data for peptide only (Cytcox and KH9) have been added to show the residual weights of these samples.

Supplementary Fig. 5: Source image for DNA binding quantification in Fig. 2e

Supplementary Fig. 6: Dynamic light scattering (DLS) analysis histograms for the estimation of particle size of the SWNT NCs.

Supplementary Fig. 10: Higher magnification images for Fig. 5b.

Supplementary Fig. 12: Source image for GFP expression in Fig. 5c.

Supplementary Fig. 13: Source image for GFP expression in Fig. 5d.

Supplementary Fig. 15: Sequencing of the PCR products from the 5' and 3' arms of the integrated pAtMTTF1 construct with primers spanning the integration site on both arms confirmed integration at the expected positions within the mitochondrial genome. Raw sequencing data have also been included as Supplementary Data 1 and 2.

Supplementary Fig. 20: Increased root growth in *A. thaliana* confirmed to be dependent on pAtMTTF pDNA integration and *SUL1* expression, by comparison with a negative expression plasmid with the *SUL1* expression cassette removed, which showed no increase in root growth.

Supplementary Fig. 21: Effect on *A. thaliana* by the integration of pAtMTTF pDNA into the mitochondria genome on photosynthetic performance, mitochondrial respiration, and folate quantification.

Supplementary Fig. 22: Plasmid maps of the reporter constructs pAtMTTF and pDONRCox2RLuc are now included.

Supplementary Table 1. List of PCR primers used for genotyping experiments in Fig. 6 and Supplementary Fig. 15 are included.

Supplementary Data 1 and 2 files containing the raw sequence reads from genotyped PCR products from the 5' and 3' arms of pAtMTTF1 transformants 7 days after infiltration with SWNT-PM-CytKH9/pDNA (Fig. 6 and Supplementary Fig. 15) have also been added.

Reviewer #1:

The manuscript describes the transformation of mitochondria (with protein and nucleic acid) using a carbon nanotube approach. This is a useful and novel approach, and with some significance as plant mitochondria remain one of the challenges to transform. While overall I am convinced by the material presented to transform mitochondria, a significant increase in efficiency on previous reports I do have some clarifications.

Comment #1

While it is great to show mitochondria are transformed and that the DNA was integrated, is it inherited into the next generation – this would really show transformation compared to transient transformation. Moreover given the almost universal maternal inheritance of mitochondria, reciprocal crosses would be needed to carry out to show that it is a maternally inherited trait.

Response

Stable integration of DNA in the mitochondria that can be inherited into the next generation remains challenging due to low delivery efficiencies into plant mitochondria as well as the instability of the plant mitochondrial genomes and the lack of strong selection markers that can be maintained during the development of the adult plant after zygote formation. In this study, we have successfully tackled the first problem by the use of peptide conjugated SWNT NCs that has sufficient delivery and expression efficiency for genome integration and expression with an observable phenotype. Furthermore, we have augmented the characterization of this phenotype with additional experiments including mitochondrial and photosynthetic measurements with folate quantification to demonstrate the potential of this system in metabolic engineering (Supplementary Fig. 21). Nevertheless, without the development of a strong selection system that can be used to tackle the instability of the mitochondrial genome, stable and inheritable transformation that can be demonstrated through maternal inheritance of the next generation remains challenging to study. This manuscript is focused on the development of a new and flexible method of DNA delivery using SWNT and peptides that would be an important first step with other applications including delivery to other organelles.

We have added this information into the conclusion section (pg. 18) of the manuscript to highlight the need of such a selection system to enable follow up studies in mitochondrial transformation and complementation.

(pg. 18)

However, organelle-specific delivery methods within plants are still lacking due to uptake limitations and relatively low delivery efficiency, and is further compounded for mitochondria targeted transformation due to the instability of the plant mitochondrial genomes and the lack of strong selection markers that can be maintained during the development of the adult plant after zygote formation. In this study, we have successfully tackled the first of these problems by the use of peptide conjugated SWNT NCs with sufficient delivery and expression efficiency for genome integration and expression with an

observable root phenotype. Further development of such a selection system would enable follow-up studies of maternal inheritance leading to genomic engineering of mitochondrial agronomic traits.

Comment #2

I was interested in the fact that the transformed plants grow better – this is an important finding, and while it cannot be fully explained in this manuscript, cannot be left without more explanation. I am presuming the plants transformed are not limited by nutrient etc, especially folate. So why are they growing better? Some basic measurements of mitochondrial activities (and photosynthetic parameters) are warranted.

Response

In order to evaluate the physiological effects of the mitochondrial integration in more detail, we evaluated the mitochondrial respiration rates via ATP generation and photosynthetic parameters by chlorophyll fluorescence (F_v/F_m) but did not observe any significant differences between the control (pDNA and SWNT-PM-CytKH9) treated samples and the samples infiltrated with the SWNT-PM-CytKH9/pDNA (Supplementary Fig. 21). Furthermore, transformation with a negative control pDNA with the *SUL1* removed did not show any enhanced root growth, providing evidence that the phenotype is due to *SUL1* integration alone (Supplementary Fig. 20).

These results suggest that overall cellular metabolism and stress were not affected by the integrating gene. Since the gene encodes for an enzyme directly used in folate synthesis, we next evaluated folate levels from plant lysates infiltrated with the complex and found that pDNA containing the *SUL1* integration and expression had a significant effect on folate levels from samples extracted 1 day and 3 days after infiltration (Supplementary Fig. 21). This suggests that the increased root growth that we observed may be a downstream pathway effect due to the increased folate concentrations as a result from the *SUL1* expression.

The results of these experiments are summarized within the Results on pg. 17 as follows and the corresponding figures have been added to Supplementary Information.

(pg. 17)

A. thaliana seedlings treated with the SWNT-PM-CytKH9/pAtMTTF1 complex also showed greater root growth (Fig. 6d and Supplementary Figs. 16-19) than seedlings treated with only pAtMTTF1 or SWNT-PM-CytKH9, as quantified by the relative root growth in area over the course of 1 week after transformation (Fig. 6d and e). No significant difference was observed between the DNA-only or SWNT-PM-CytKH9/pDNA complex treated seedlings when a negative control construct with the *SUL1* and its promoter and terminator removed (Supplementary Fig. 20).

Chlorophyll fluorescence and mitochondrial respiration measurements did not show any significant differences upon SWNT NCs infiltration, suggesting that the overall increase in root area was not due to overall metabolic changes and that the plants were not stressed by the SWNT NCs introduction (Supplementary Fig. 21a-b). We then quantified intracellular folate concentrations that showed significantly increased folate levels relative to the pDNA

only and SWNT-PM-CytKH9/pDNA (-*SUL1*) infiltrated samples at days 1 and 3 post-infiltration (Supplementary Fig. 21c). We hypothesize that the expression of *SUL1* from the transformed construct led to the observed increased folate levels, thus increasing the growth rate observed within the roots.

Supplementary Fig. 20. Effect on *A. thaliana* root growth by infiltration of SWNT-PM-CytKH9/pDNA with and without the *SUL1* insert.

A. thaliana root growth after infiltration with SWNT-PM-CytKH9 complexed with pAtMTTF1 pDNA containing the *SUL1* insert (+*SUL1*) and the negative control with the -insert removed (-*SUL1*) relative to their respective pDNA only control. Significant differences were observed between the relative root areas (n=18 seedlings) on Day 3, 5, and 7.

Supplementary Fig. 21. Photosynthetic performance, mitochondrial respiration activities, and intracellular folate concentrations of *A. thaliana* infiltrated with SWNT NCs and its pDNA complex. (a) Representative samples of chlorophyll fluorescence (F_v/F_m) measurements of seedlings at Day 1, 3, and 7 after infiltration with pDNA (+*SUL1*) only, SWNT-PM-CytKH9 with pDNA (+*SUL1* or -*SUL1*) showed no significant differences between the measured samples. (b) Quantification of chlorophyll fluorescence (F_v/F_m) of samples from (a) across 2 plates per condition containing a minimum of 10 leaf areas each using the FluorCam software (Version 7) showed no significant difference between the measured samples. Average values of F_v/F_m ranging from 0.75-0.80 for all sample suggest low levels of stress in plants even after SWNT NC treatment. (c) Mitochondrial respiration as quantified by ATP generation from isolated mitochondria from *A. thaliana* showed no significant differences in respiration or mitochondrial stress upon infiltration with SWNT-NCs and its complex or DNA only. (d) Folic acid quantification of *A. thaliana* infiltrated with SWNT NCs or DNA only show a significant increase in folic acid levels at Days 1 and Day 3 post-infiltration for SWNT-PM-CytKH9/pDNA (+*SUL1*) treated samples relative to the pDNA only and SWNT-PM-CytKH9 samples and at Day 1 relative to SWNT-PM-CytKH9/pDNA (-*SUL1*).

Comment #3

A number of mutants in nuclear genes encoding mitochondrial proteins or mitochondrial located genes encoding proteins that result in male infertility have been reported in

Arabidopsis. The complementation of at least one of these by expressing in mitochondria would show the utility of this system.

Response

We agree that complementation of mitochondrial mutants is the next important step in demonstrating the utility of this system for mitochondrial transformation. However, to achieve stable and homologous expression of transgenes in mitochondria, a well-developed selection marker system in mitochondria would be required to tackle problems associated with the instability of mitochondrial genome. We have begun efforts into developing a selection system by expressing the *SUL1* sulfadiazine resistance gene using genetic integration in this work, but sufficient stable integration of transgenes has still not yet been achieved. Thus, we believe these complementation experiments would be beyond the scope of the current work. We hope the reviewer understands this technical issue and our current focus in this work.

Comment #4

In summary as presented the manuscript reports a substantial increase in efficiency for plant mitochondrial transformation. However, as this is not the first time plant mitochondria have been transformed, some useful utilizations of this system are necessary to show it can be inherited and address some of the issues that the authors raised in the introduction – otherwise it would be a rather specialised system with limited uses.

Response

Although this method is not the first time plant mitochondria have been transformed, this is the first time that a SWNT-based delivery system has been used to deliver pDNA into plant mitochondria. Importantly, this system shows a 30-fold increase in expression efficiency compared with previous peptide/pDNA based systems and we show for the sufficient integration of the transgene with an observable root growth phenotype and elevated folic acid levels as a result of this integration, suggesting there is strong potential for metabolic engineering in mitochondrial transformation. Furthermore, the flexibility of the SWNT NCs developed in this manuscript allows facile customization of the conjugated peptides to develop delivery systems to other organelles carrying different cargoes.

We have highlighted these advances within the introduction (pg. 4) as follows.

(pg. 4)

Our results also showed that the expression of a folate metabolism-related gene conferred increased root growth and folate levels with potential for metabolic engineering, including the development of a selection marker system for stable transformation of plant mitochondria.

Reviewer #2:

Results summary:

The work by Lau et al uses polymer coated carbon nanotubes modified with peptides to enable plasmid delivery into the mitochondria of plants. The team uses different reporters to validate their platform and are able to quantify protein expression with western blots and fluorescence microscopy. The concept and advance is very high in its significance for mitochondria transformation in plants, which is likely unprecedented. However, given its potential very high impact, there are several points that should be addressed by the authors prior to publication as noted below.

Main points:

Comment #1

Colorimetric assays such as Evans blue relies on absorbance and emission of dyes. It is known that SWNT have very broad absorption and emission spectra that overlap with those of dyes. qPCR of plant stress genes should be performed for molecular-level validation of biocompatibility for the SWNT-PM-Peptide/pDNA complexes.

Response

We further confirmed the stress upon infiltration with the SWNT complexes by measurement of mitochondrial respiration and photosynthetic efficiency by ATP production and chlorophyll fluorescence, respectively (Supplementary Fig. 21). Both of these parameters did not show any significant decrease in efficiency, suggesting that toxicity and stress effects by the SWNT were relatively low. Furthermore, root growth was relatively unaffected by infiltration with the SWNT-PM-CytKH9 compared with those infiltrated with the pDNA only, also indicating that there are relatively low toxicity and stress effects by infiltration of the SWNT.

The results of these experiments are summarized within the Results (pg. 17) as follows within the manuscript and the corresponding figures have been added to the Supplementary Information.

(pg. 17)

Chlorophyll fluorescence and mitochondrial respiration measurements did not show any significant differences upon SWNT NCs infiltration, suggesting that the overall increase in root area was not due to overall metabolic changes and that the plants were not stressed by the SWNT NCs introduction (Supplementary Fig. 21a-b). We then quantified intracellular folate concentrations that showed significantly increased folate levels relative to the pDNA only and SWNT-PM-CytKH9/pDNA (-*SUL1*) infiltrated samples at days 1 and 3 post-infiltration (Supplementary Fig. 21c). We hypothesize that the expression of *SUL1* from the transformed construct led to observed increased folate levels, thus increasing the growth rate observed within the roots.

Supplementary Fig. 21. Photosynthetic performance, mitochondrial respiration activities, and intracellular folate concentrations of *A. thaliana* infiltrated with SWNT NCs and its pDNA complex. (a) Representative samples of chlorophyll fluorescence (F_v/F_m) measurements of seedlings at Day 1, 3, and 7 after infiltration with pDNA (+*SUL1*) only, SWNT-PM-CytKH9 with pDNA (+*SUL1* or -*SUL1*) showed no significant differences between the measured samples. (b) Quantification of chlorophyll fluorescence (F_v/F_m) of samples from (a) across 2 plates per condition containing a minimum of 10 leaf areas each using the FluorCam software (Version 7) showed no significant difference between the measured samples. Average values of F_v/F_m ranging from 0.75-0.80 for all sample suggest low levels of stress in plants even after SWNT NC treatment. (c) Mitochondrial respiration as quantified by ATP generation from isolated mitochondria from *A. thaliana* showed no significant differences in respiration or mitochondrial stress upon infiltration with SWNT-NCs and its complex or DNA only. (d) Folic acid quantification of *A. thaliana* infiltrated with SWNT NCs or DNA only show a significant increase in folic acid levels at Days 1 and Day 3 post-infiltration for SWNT-PM-CytKH9/pDNA (+*SUL1*) treated samples relative to the pDNA only and SWNT-PM-CytKH9 samples and at Day 1 relative to SWNT-PM-CytKH9/pDNA (-*SUL1*).

Comment #2

Authors claim that the Cytcox peptide confers specific mitochondrial delivery while the data suggest that SWNT location is unspecific. SWNT-PM-KH9 in SFig6 are shown to be

localized to mitochondria while there is no targeting peptide in this material (so localization should not be specific). Authors show that SWNT-PM-CytKH9 are able to deliver pDNA to both mitochondria and nucleus for expression. From the data they present, organelle-specific expression is given by the promoter in the pDNA, not by the Cytcox peptide, I think the text should explicitly reflect that. Their data clearly show that the combination of Cytcox and KH9 is what gives the highest expression, making CytKH9 a *novel* and very versatile peptide for delivery, not restricted to mitochondria. Because authors already have plasmids with chloroplast promoters, I would suggest that they try to deliver them and see if they detect expression.

Response

We agree that the delivery of the pDNA by SWNT-PM-CytKH9 were not organelle specific and the organelle-specific expression was conferred by the promoter in the pDNA. We have clarified this within the main text (pg. 14-15) as follows:

(pg. 14-15)

We note that the SWNT NCs likely had a degree on non-specificity and could localize within the nucleus in addition to the mitochondria as evidenced by the expression with the 35S promoter pDNA and that the observed organelle specific expression was from the promoters used in the respective constructs.

We also attempted to deliver the pDNA using SWNT-PM-CytKH9 for chloroplast expression of the same *Renilla Luciferase* expression construct with the *psbA* promoter but did not observe significant expression. Please see the luciferase expression graph below.

Comment #3

Regarding the claim of HDR in mitochondria, the PCR should be better explained and the sequencing data should be provided in the SI. Moreover, if there is HDR, the bottom panel of figure 6b should show a larger band corresponding to the locus with the introduced genes. Also, to sustain the claim that increased folate results in root growth, I would suggest that they show root folate quantification. Alternatively, a good negative control would be to deliver pDNA with only the GFP reporter lacking the *sul1* gene.

Response

The description of the PCR has been expanded in the methods section (pg. 26) as follows:

(pg. 26)

PCR genotyping of transformed plant tissues

Total DNA was extracted from seedlings with the DNeasy Plant Mini Kit (QIAGEN). Primers were designed to amplify the wild-type mitochondrial genomic locus containing the integration sites for pAtMTTF1, outside of the homology arms. Primers were also designed for detecting the presence of the recombined construct, such that one primer binds outside of the homology arm and the other primer binds within the coding sequence of a gene in the construct, spanning either the left or right junction. Primers were also designed for amplifying the integrated construct, spanning either the left or right homology arm of pAtMTTF1. PCR was performed using PrimeSTAR GXL polymerase (Takara), with an annealing temperature of 64°C and 35 amplification cycles for the left and right arm reactions, and 60 °C and 30 cycles for the wild-type reaction. DNA sequencing of the corresponding extracted bands was performed using the respective primers in Fig. 6a and included as Supplementary Data 1-2.

We believe that HDR is occurring, but at a frequency too low to be detected by PCR using the primer pair in the bottom panel of Figure 6b. Thus, the only PCR product visible in that reaction would be amplified from the wild-type locus in the mitochondrial genome.

Sequencing data has been added to Supporting Fig. 15, and the raw sequencing data has also been included as Supplementary Data 1 and 2. The sequencing reads span the junction between the mitochondrial genome and the homologous sequence from the HDR donor plasmid, showing that recombination occurred at the expected location.

Supplementary Fig. 15. Sequencing of the genotyping PCR products from the 5' and 3' arms of the integrated pAtMTTF1 construct. Sequencing reads spanning the junction between the mitochondrial genome and the homologous sequence from the donor plasmid for recombination on both arms confirmed integration at the expected positions within the

mitochondrial genome. Primers 1 and 4 refer to the primers labelled in Fig. 6a. Raw sequence reads can be found in the Supplementary Data 1 and 2.

We cloned the pDNA with the *SUL1* gene and associated promoter and terminator deleted and repeated the root growth experiment with this negative control as mentioned (Supplementary Fig. 20). There was increased root growth only when pDNA containing the *SUL1* gene was delivered with SWNT, but not when the negative control was delivered with SWNT as compared to the respective pDNA only controls. We also evaluated folate levels from plant lysates and found that pDNA containing the *SUL1* integration and expression had a significant effect on folate levels from samples extracted 1 day and 3 days after infiltration (Supplementary Fig. 21). This suggests that the increased root growth that we observed may be a downstream pathway effect due to the increased folate concentrations as a result from the *SUL1* expression.

These results are now included within the main text (pg. 17).

(pg. 17)

No significant difference was observed between the DNA-only or SWNT-PM-CytKH9/pDNA complex treated seedlings when a negative control construct with the *SUL1* and its promoter and terminator removed (Supplementary Fig. 20).

Supplementary Fig. 20. Effect on *A. thaliana* root growth by infiltration of SWNT-PM-CytKH9/pDNA with and without the *SUL1* insert.

A. thaliana root growth after infiltration with SWNT-PM-CytKH9 complexed with pAtMTTF1 pDNA containing the *SUL1* insert (+*SUL1*) and the negative control with the -insert removed (-*SUL1*) relative to their respective pDNA only control. Significant differences were observed between the relative root areas (n=18 seedlings) on Day 3, 5, and 7.

Supplementary Fig. 21. Photosynthetic performance, mitochondrial respiration activities, and intracellular folate concentrations of *A. thaliana* infiltrated with SWNT NCs and its pDNA complex. (d) Folic acid quantification of *A. thaliana* infiltrated with SWNT NCs or DNA only show a significant increase in folic acid levels at Days 1 and Day 3 post-infiltration for SWNT-PM-CytKH9/pDNA (+*SULI*) treated samples relative to the pDNA only and SWNT-PM-CytKH9 samples and at Day 1 relative to SWNT-PM-CytKH9/pDNA (-*SULI*).

Comment #4

Based on the provided FE-SEM micrographs, there are no SWNTs visible. The red arrows are pointing to features we see prolifically in plant leaf tissue EM of inclusion bodies. SWNT and peptides are carbon based and are therefore highly unlikely to be observable inside plant cells except in the vacuole. I suggest removing the SEM micrographs as they do not help support the claim of internalized SWNT. To this end, the resolution of confocal Raman (unspecified) is unlikely to resolve SWNT present in the mitochondria against those present in the cell or on the surface of the mitochondria or more broadly in plant tissue. Also please clarify why the baseline Raman intensity for the different SWNT treatments varies so greatly.

Response

Additional experiments using fluorescently labelled SWNT-PM-CytKH9 and wide area Raman mapping of the relevant SWNT G and G' peaks have been added to support the localization of SWNT within the isolated mitochondria upon infiltration (Fig. 3a-b). Raman measurements were shifted in the y-axis for clarity, but have been replotted to the original positions in the revised figure (Fig. 3c). We apologize for any confusion in the previous graphs.

The results have been incorporated into the Results of the main text (pg. 9-11) as follows:

(pg. 9-11)

To investigate the delivery capabilities of the SWNT NCs to plant mitochondria, we prepared fluorescently labelled SWNT-PM-CytKH9 using DyLight 488-conjugated KH9Cys and examined its localization within the root cells of *A. thaliana* upon vacuum/pressure infiltration at 0.08 MPa for 60 seconds. In the cells infiltrated with the labelled SWNT-PM-CytKH9, clear colocalization between DyLight 488-labelled SWNT-PM-CytKH9 and MitoTracker-stained mitochondria could be observed in the samples infiltrated with the labelled SWNT-PM-CytKH9 as well as localization of the labelled SWNT-PM-CytKH9 within the cytosol of the root cells (Fig. 3a, upper panels). Conversely, most of the labelled SWNT-PM-CytKH9 samples remained on the surface of the root without vacuum infiltration (Fig. 3a, lower panels).

Next, we analyzed isolated mitochondria from *A. thaliana* infiltrated with SWNT NCs by confocal Raman microscopy (Fig. 3b-d). Raman mapping of the G band peak at 1590 cm^{-1} showed clear colocalization of SWNT signal with the isolated mitochondria from seedlings infiltrated with SWNT-PM-KH9 and SWNT-PM-CytKH9 (Fig. 3b) that is not present in the samples infiltrated with SWNT-PM. Similar levels of colocalization was also observed in samples infiltrated with SWNT-PM-CytKH9 but those infiltrated with SWNT-PM-KH9 and

SWNT-PM exhibited considerably less G-band signal, suggesting that the Cytcox peptide played a role in directing the SWNT NCs into the mitochondria. To further quantify the effect of the peptide conjugation, we collected Raman spectra over an area of isolated mitochondria treated with the SWNT NCs and the overlay of the averaged Raman spectra (Fig. 3c) showed that similar characteristic G and G' bands at 1590 cm^{-1} and 2640 cm^{-1} (Fig. 3c and d). Quantification of the normalized 1590 cm^{-1} (Fig. 3d) and 2640 cm^{-1} (Supplementary Fig. 8) peak heights showed that all three SWNT NCs were detected within the isolated mitochondria. In particular, the strongest Raman signals were detected from the samples infiltrated with SWNT-PM-CytKH9 with an approximately 10-fold increase in normalized G-band intensity relative to the samples infiltrated with SWNT-PM and SWNT-PM-KH9, suggesting that the dually functionalized SWNT was delivered most efficiently into the mitochondria (Fig. 3d and Supplementary Fig. 8). Taken together with the fluorescently-labelled SWNT results, these findings show that the SWNT NCs can be localized within plant mitochondria, with the Cytcox peptide conferring increased mitochondrial targeting.

Fig. 3. CLSM of fluorescently labelled SWNT NCs and Raman microscopy of mitochondria in *A. thaliana* upon introduction of SWNT NCs.

(a) Fluorescently-labelled (DyLight488) SWNT-PM-CytKH9 were detected within MitoTracker-labelled mitochondria and cytosol upon vacuum infiltration (upper panels) and on the surface of the root cells without vacuum infiltration (lower panels) when introduced into intact *A. thaliana* seedlings. White arrows indicate colocalization of the DyLight488-labelled SWNT-PM-CytKH9 and MitoTracker-labelled mitochondria. **(b)** Raman microscopy mapping of the 1490 cm^{-1} G-band peak taken from isolated mitochondria isolated from *A. thaliana* 18 h after infiltration showed colocalization of SWNT signal with the isolated mitochondria in the samples infiltrated with SWNT-PM-CytKH9 and SWNT-PM-KH9. **(c)**

Representative Raman spectra taken from mitochondria isolated from *A. thaliana* 18 h after infiltration show characteristic G (1590 cm⁻¹) and G' (2640 cm⁻¹) bands from the SWNTs. **(d)** Quantified normalized G-band peak (1590 cm⁻¹) intensities for isolated mitochondrial samples shown in **(b)**. Relative values from 64 individual spectra (n=64) are shown for each individual sample. ns – not statistically significant, ***P<0.001, ****P<0.0001.

Comment #5

Figure 6a-b: It seems that given the design of the primers, a PCR of the delivered pDNA complex (not necessarily the mitochondrial genome) would give rise to amplification of the same region. It may be that the PCR amplicon has amplified to a greater extent with the SWNT sample because SWNTs are able to better protect the pDNA from degradation. It might be necessary to sequence regions that begin past the left and right homology arms to determine whether authors are amplifying a region from the pDNA or a mitochondrial genomic region.

Response

We are sorry for this confusion. The original figure and description of the PCR showed the outer primers in the wrong position; they have been revised to show that the primer pairs do span past the left and right homology arms, so they show that the pDNA sequence has undergone recombination with the mitochondrial genome. The primer sequences are also included in Supplementary Table 1.

Fig. 6. Genotyping and phenotypic effects in *A. thaliana* upon infiltration with SWNT-PM-CytKH9/pAtMTTF1 pDNA complexes.

(a) Design of a plasmid DNA construct (pAtMTTF1) for integration into the mitochondrial genome of *A. thaliana*. *SUL1* encodes dihydropteroate synthase type-2, which was previously used as a selection marker for mitochondrial transformation³⁶.

Sequencing data of the genotyping PCR products has also been added (Supplementary Fig. 15 and Supplementary Data 1 and 2). The sequencing reads span the junction between the mitochondrial genome and the homologous sequence from the HDR donor plasmid, showing that recombination occurred at the expected location.

This information is clarified within the main text as well (pg. 16).

(pg. 16)

PCR genotyping of tissue samples 7 days after infiltration showed that the exogenous sequence was successfully integrated into the mitochondrial genome (Fig. 6b). DNA integration events were observed in all samples treated with SWNT-PM-CytKH9/pAtMTTF1 complexes, as indicated by positive genotyping PCR products for both the left and right homology arms (Fig. 6b). These PCR products span the junctions between the mitochondrial genome and the homologous sequence from the donor plasmid, and thus would only be

amplified in the case of successful recombination. (Fig. 6b). In contrast, positive PCR products were detected faintly in only one sample out of six treated with pAtMTTF1 alone. Sequencing of the PCR products from the 5' and 3' arms of the construct confirmed integration at the expected positions within the mitochondrial genome (Supplementary Fig. 15).

Supplementary Fig. 15. Sequencing of the genotyping PCR products from the 5' and 3' arms of the integrated pAtMTTF1 construct. Sequencing reads spanning the junction between the mitochondrial genome and the homologous sequence from the donor plasmid for recombination on both arms confirmed integration at the expected positions within the mitochondrial genome. Primers 1 and 4 refer to the primers labelled in Fig. 6a. Raw sequence reads can be found in the Supplementary Data 1 and 2.

Minor points:

Comment #6

Root expression is particularly exciting. However I am unclear how root expression assays were done: were the roots vacuum infiltrated? Were these roots from plants that were leaf infiltrated? Hydroponic assays? Line 237 simply says “after infiltration” but it is unclear what was infiltrated and when.

Response

We apologize for this unclear description. Whole plant seedlings (7 day old) were infiltrated with the SWNT-PM-Peptide/pDNA solutions and were plated on MS media plates after infiltration. Root lengths and all other characterization were determined as days after initial infiltration. We have clarified this within the main text as follows:

(pg. 11)

“To investigate the delivery efficiency of the pDNA within plants, we evaluated the expression of a pDNA encoding a *Renilla* luciferase reporter construct (*RLuc*) upon the same vacuum/pressure infiltration conditions (0.08 MPa for 60 s) in the previous experiment within whole *A. thaliana* seedlings.”

(pg. 24)

“Seven-day-old *A. thaliana* seedlings were used to assess the delivery of pDNA and its expression and integration in *A. thaliana*. Vacuum/pressure infiltration was performed by incubation of whole seedlings (10 μ L solution per seedling) in the respective solution and subjected to 0.08 MPa vacuum followed by pressure for 60 seconds each. The seedlings were allowed to recover in the solution for 1 h at room temperature before being plated on 0.5x MS medium plates. For root growth measurements, the plates were placed at an orientation of approximately 75°. Seedlings were allowed to grow under 16-/8-h light/dark periods at 22°C for 1–7 days.”

Comment #7

Authors show that loading of the plasmid generates a closer to neutral SWNT-plasmid suspension. Authors should test whether a larger plasmid could be adsorbed to the SWNT (not suggesting all of the study be redone, just the gel shift assays and colloidal stability tests).

Response

The integrating plasmid DNA containing the *SUL1* was relatively large (10 kb) compared to the transiently expressed plasmid (5 kb). Gel shift assays of this plasmid with the respective SWNT samples are included in Supplementary Fig. 14.

Supplementary Fig. 14. pAtMTTF pDNA complexation with SWNT-PM-CytKH9.

Quantification of gel shift electrophoresis mobility assays showing the amount of residual uncomplexed pAtMTTF pDNA (10 kb) at different SWNT-PM-Peptide NPs/pDNA ratios.

This information was also made more explicit within the main text (pg. 15) as:

(pg. 15)

SWNT-PM-CytKH9 complexed similarly with the pAtMTTF1 pDNA (10 kb) compared to the previously used pDONR pDNA (5 kb), suggesting that the SWNT NCs are able to complex with larger plasmids (Supplementary Fig. 14).

Comment #8

Figure 4 figure nor caption mentions what the biological substrate is, please mention it is leaves (I am assuming?)

Response

The *Renilla* luciferase assays were evaluated using whole seedling soluble lysate fractions as performed in previous studies. The Evans blue test for toxicity was evaluated using intact seedlings. This information is now clarified within the Fig. 4 caption.

“(b) Transfection efficiency of the SWNT-PM-Peptide and pDNA was evaluated 18 h after infiltration using a *Renilla* luciferase reporter construct under the control of mitochondria-specific (Cox2) and nuclear (35S) promoters using whole seedling soluble lysate fractions. Data from 6 biologically independent samples (n = 6) are shown. (c) Evans blue assays for intact *A. thaliana* seedlings 18 h post infiltration with each respective NC, normalized to the absorbance of a boiled sample. *A. thaliana* infiltrated with DNA alone was used as a control.”

Comment #9

How come 200 ng of pDNA was loaded on the SWNT-PM-peptide complex? Please include reasoning in line 203.

Response

200 ng pDNA was loaded based on the gel-shift assays and the binding capacity of the SWNT-PM-CytKH9.

This information has been added into the text (pg. 11) as “Based on DNA binding capacity of SWNT-PM-CytKH9, SWNT-PM-Peptide (400 µg) was complexed with 200 ng of pDNA for infiltration experiments (Fig. 2e and f)”.

Comment #10

Line 48: Carbon nanotubes can be made from graphite but are not made of graphite.

Response

This is now corrected within the text (pg. 3) as “Carbon nanotubes (CNTs) are cylindrical-tubule structures made from graphite with exceptional physical properties that have been harnessed in biological applications such as drug delivery and biosensors^{14,15}.”

Comment #11

Figure 2a-c: It would be helpful for visual comparison if you could standardize the size of the scale bars across all samples, and also if the scale bar were smaller in size (100 nm or smaller).

Response

The scale bars have been standardized at 200 nm across the samples and Fig. 2a has been resized to match the other figures. The caption of Fig. 2a has also been modified to reflect this.

Fig. 2 (a) FE-SEM micrographs of (a) SWNT-PM, (b) SWNT-PM-CytKH9, and (c) SWNT-PM-CytKH9/pDNA show typical SWNT morphology after excess peptide has been removed by dialysis. Scale bars represent 200 nm.

Comment #12

Figure 2a-c: From the images it looks like the width of the SWNTs are quite large, quite a bit larger than unmodified SWNTs. Do you know what the average width is, especially across samples?

Response

Based on the SEM images and AFM results, we expect the coating on the surface of the SWNT to be relatively uniform, so we expect the increases in the observed AFM heights (~2 nm) to be also reflective of the different in widths observed between the unmodified and modified SWNTs. This is clarified within the main text (pg. 8) as follows:

(pg. 8)

The profiled AFM heights were relatively uniform for the measured samples (Fig. 2j) and increased from 0.81 ± 0.26 nm to 2.14 ± 0.24 nm upon polymer coating, giving a polymer layer thickness of approximately 1.3 nm which agrees with our previous results²⁴.

Comment #13

Figure 2g: Please include zeta potentials of the different SWNT samples after DNA has been loaded, I imagine the zeta potential would be highly dependent on amount of DNA.

Response

Zeta potentials in the presence of pDNA are now included in Fig. 2g. We have also added the following information into the main text (pg. 8):

(pg. 8)

This was also reflected in the change in zeta potential upon complexation with pDNA, where SWNT-PM-KH9 saw the greatest change (-22.2 mV) when compared with SWNT-PM-CytKH9 (-4.8 mV), and SWNT-PM-Cytcox (+2.4 mV).

Fig. 2 (g) Zeta potentials of SWNT-PM-Peptide with different functional peptides **in the absence and presence of pDNA**. Data are represented as the mean \pm standard deviation values ($n = 3$).

Comment #14

Supplementary Fig. 2: From what I understand, SWNT-PM-CytKH9 is derived from SWNT-PM, so it would be logical that mass % remaining at 500C (that is, what we attribute to pure SWNT) would be lower for the SWNT-PM-CytKH9 sample than SWNT-PM. Why is this not so?

Response

Changes in solubility of the peptide conjugated sample (SWNT-PM-CytKH9) as well as ashing of the peptides themselves may have resulted in a higher residual weight at higher temperatures during TGA analysis. To confirm this, we performed TGA analysis on peptide samples (CytcoxCys and KH9Cys) and both peptides showed similar residual weight at the highest temperature, with KH9 having higher residual weight than the SWNT-PM sample. These results have been included in Supplementary Fig. 2 and the main text (pg. 6) as follows.

(pg. 6)

Thermogravimetric analysis (TGA) of the SWNT NCs was used to compare the mass loss profiles from 30 to 500°C (Supplementary Fig. 2) and evaluate their composition. Pristine SWNTs had minimal weight loss during pyrolysis, as expected for CNTs. SWNT-PM and SWNT-PM-CytKH9 showed a large mass loss at approximately 300°C, which was completed at approximately 420°C. **This agrees well with residual amounts after pyrolysis of the maleimide and peptide samples and suggests that it corresponds to the adsorbed polymer layer and conjugated peptides.** The residual weights at 500°C for SWNT-PM (22%) and SWNT-PM-CytKH9 (33%) suggested that the weight ratio of polymer to SWNT to peptide is approximately 10:3:2 in SWNT-PM-CytKH9.

Supplementary Fig. 2. TGA analysis of the SWNT NCs.

TGA analysis curves for the SWNT NCs and unconjugated peptides showing their mass loss profile during pyrolysis.

Comment #15

Please number your Supplementary Figures in order of appearance (10 comes before 9 in the manuscript).

Response

The order of Supplementary Figures has been corrected to the order of appearance.

Comment #16

Figure 5b: Please provide scale bars on your images. Also, it would be great to have some higher magnification images available.

Response

Scale bars have been added and higher magnification images are now included within Supplementary Figure 10.

Fig. 5. (b) Representative confocal laser scanning microscopy images of *A. thaliana* root cells 18 h post infiltration with SWNT-PM-CytKH9/pDNA complexes containing *pDONR-35S-GFP* or *pDONR-Cox2-GFP* reporter constructs for nuclear or mitochondrial expression, respectively. Mitochondria were stained with MitoTracker Red (CMXRos), and colocalization analysis was performed on GFP expression and MitoTracker signals. Scale bars represent 20 μm .

Supplementary Fig 10. Confocal laser scanning microscopy images of *A. thaliana* root cells 18 h post infiltration with SWNT-PM-CytKH9/pDNA complexes. Higher magnification images of *A. thaliana* infiltrated with SWNT-PM-CytKH9/pDNA shown in Fig. 5b. Mitochondria were stained with MitoTracker Red (CMXRos), and colocalization analysis was performed on GFP expression and MitoTracker signals. Scale bars represent 20 μ m.

Comment #17

Throughout the entire manuscript, authors should state prior to every result how many days-post-infiltration (dpi) their results were resultant from.

Response

The number of days-post-infiltration (dpi) has been explicitly added to the results without this information in the main text.

(pg. 12)

“Both the 35S ($19,000 \pm 2700$ RLU/mg protein) and Cox2 ($34,000 \pm 4500$ RLU/mg protein) promoters showed the highest levels of luciferase activity for the samples infiltrated with SWNT-PM-CytKH9/pDNA (Fig. 4b) after overnight incubation of 18 h.”

(pg. 16)

“PCR genotyping of tissue samples 7 days after infiltration showed that the exogenous sequence was successfully integrated into the mitochondrial genome (Fig. 6b).”

Comment #18

It is unclear from initial reading of the text and figure 1 whether the peptides are attached to the SWNT covalently or noncovalently. I suggest reworking the text and figure accordingly. I.e. line 74-27 refers to prior publications but this is critical information that needs to be included in this manuscript (it isn't until line 327 that authors mention that the polymer is noncovalently coated on SWNT).

Response

Information regarding the noncovalent nature of the maleimide methacrylate layer adsorbed on the surface of the SWNT and the covalently linked peptides have been added to the main text. Additional information regarding the peptides have also been added to the introduction and results sections as follows:

(pg. 4)

The maleimide methacrylate layer is non-covalently adsorbed on the surface of the SWNT using micelle-polymerization and can be covalently conjugated with thiol-containing moieties. The Cytochrome peptide contains the mitochondrial targeting presequence that has been previously characterized and used for mitochondrial pDNA delivery and expression within plant and animal cells²⁶. KH9 is a cationic peptide that facilitates electrostatic interactions with anionic pDNA for binding.

(pg. 5)

Furan-protected maleimide was polymerized with polyethylene glycol (PEG) methacrylate as a non-covalently bound polymer layer on dispersed SWNTs (Supplementary Fig. 1a). Upon deprotection, the maleimide group can be covalently modified with thiol-containing molecules via Michael addition (Supplementary Fig. 1b and c). To target SWNT-PM for DNA delivery into mitochondria, we covalently conjugated two distinct Cys-containing functional peptides to yield SWNT-PM-CytKH9 (Fig. 1a).

Comment #19

In general, the authors suggest that a mitochondrial localization sequence tethered to these relatively large nanostructures enables trafficking into the mitochondria. Are the entire SWNT structures, which can apparently be on the same size magnitude as mitochondrion themselves be trafficked in their entirety into these organelles? Or are the SWNT constructs somehow shedding these peptides and pDNA cargo in the cell? It would be good for authors to speculate, in addition to providing information on the average length and polydispersity of the SWNT used.

Response

The lengths of the conjugated SWNT NCs introduced into the plants averaged approximately 200-300 nm based on DLS measurements with PDI around 0.4-0.5. The overall images taken using AFM also support these size and distribution. Relative to the size of the plant mitochondria (1-3 μm), it would be feasible that the entire SWNT complex is able to enter, and our experimental data using fluorescently labelled SWNT and confocal Raman microscopy support this. However, based on the resolution achievable with these techniques we cannot rule out the fact that the SWNT NCs could be embedded on the surface or only

partially translocated into the mitochondria with the cargo released. This explanation has been added to the main text (pg. 8) and Supplementary Fig. 6 as follows:

(pg. 8)

Long strands of SWNT-PM could be observed on the graphite substrate and on average ranged from 200–1000 nm that corresponded well with the average lengths (200-400 nm and PDI: 0.4-0.55) observed by dynamic light scattering (Supplementary Fig. 6).

Supplementary Fig. 6. Dynamic light scattering measurements of SWNT-PM-Peptide NCs. DLS measurements of particle size showed similar results with those observed using AFM.

Comment #20

Line 204. It is surprising that a human derived COX2 mitochondrial promoter is active in

plant mitochondria. Are there literature to support this inter-kingdom adaptability? Are there any possible sources of non-specific transcription from this promoter? Additionally, the COX2 promoter elements are quite complex in their organization. How was the exact sequence to use determined? Additionally, could the authors explain the use of tNOS in their mitochondrial plasmid construct? It is surprising that a terminator used in nucleus expression is effective in this scenario as well.

Response

In the pAtMTTF1 construct, GFP is under the control of an *Arabidopsis thaliana* COX2 promoter, and SUL1 is under the control of a *Saccharomyces cerevisiae* COX2 promoter. In both of these cases, COX2 refers to the cytochrome *c* oxidase subunit 2 in the mitochondrial genome. These promoter sequences were determined by taking approximately 1 kilobase upstream from the respective translation start sites in the mitochondrial genomes. In the pDONR-cox2p-RLuc and GFP constructs, the promoter is from *S. cerevisiae* COX2. There is no terminator sequence according to the sequence map, and this has been removed from the construct diagrams in Figs. 4 and 5. We apologize for the initial error in the diagram label.

Previous studies have demonstrated that deletion of the untranslated terminator sequence did not affect expression of pDNA containing reporter constructs introduced within plant mitochondria, possibly due to the complex mRNA maturation and editing processes in plant mitochondria (Farré, J. and Arayaa, A.. Nucleic Acids Res. 2001 Jun 15; 29(12): 2484–2491; Dombrowski, S., Brennicke, A., Binder, S. The EMBO Journal (1997)16:5069-5076.).

Furthermore, the expression of these plasmids specifically within the mitochondria have been demonstrated in our previous work (Chuah, J. et al. Scientific Reports volume 5, 5, 7751 (2015). <https://doi.org/10.1038/srep07751>; Chuah, J. et al. *Biomacromolecules* 2016 17 (11), 3547-3557 DOI: 10.1021/acs.biomac.6b01056).

Information regarding the constructs is now included within the main text (pg. 23) as follows:

(pg. 23)

Plasmids containing the GFP and luciferase reporter constructs were previously prepared and used for expression within *A. thaliana* in previous studies (Plasmid maps are shown in Supplementary Fig. 22)^{12,13,46}. The pDONR cox2p-RLuc and GFP constructs contain the promoter from *S. cerevisiae* COX2 and the reporter construct, without a terminator sequence. Previous studies have demonstrated that deletion of the untranslated terminator sequence did not affect expression of pDNA containing reporter constructs introduced within plant mitochondria^{47,48}.

The respective plasmid maps have also been added into Supplementary Fig. 22.

Fig. 4. Transfection efficiency of the luciferase reporter construct and cytotoxicity of SWNT NCs.

(a) Reporter construct design of transient luciferase expression for the nucleus and mitochondria.

Fig. 5. GFP Expression within *A. thaliana* upon infiltration with SWNT-PM-CytKH9/pDNA complexes.

(a) Reporter construct design of transient GFP expression for the nucleus and mitochondria.

Supplementary Fig. 15. Sequencing of the genotyping PCR products from the 5' and 3' arms of the integrated pAtMTTF1 construct. Sequencing reads spanning the junction between the mitochondrial genome and the homologous sequence from the donor plasmid for recombination on both arms confirmed integration at the expected positions within the mitochondrial genome. Primers 1 and 4 refer to the primers labelled in Fig. 6a. Raw sequence reads can be found in the Extended Data files.

Supplementary Fig. 22. Plasmid maps of the reporter constructs pAtMTTF1, pDONR-Cox2Rluc, and pDONR-Cox2GFP.

Comment #21

Could authors cite literature for the statement that HDR occurs readily within plant mitochondria?

Response

The references for HDR occurring readily within plants is now included in the text.

“This construct is expected to be integrated into the mitochondrial genome via homologous recombination, which occurs readily within plant mitochondria^{49,50}.”

References

- 49. Mileshina, D., Koulintchenko, M., Konstantinov, Y. & Dietrich, A. Transfection of plant mitochondria and in organello gene integration. *Nucleic Acids Res.* **39**, e115 (2011).
- 50. Møller, I. M., Rasmusson, A. G. & Van Aken, O. Plant mitochondria – past, present and future. *Plant J.* **108**, 912–959 (2021).

Comment #22

Line 286. In the literature cited for this selective marker, it appears that the *sul1* protein was expressed by nucleus expression and then subsequently targeted to the mitochondria using a localization sequence. It was not developed for mitochondrial transformation. Is the transgene used in this experiment the same? If so, could there be

erroneous nuclear expression and subsequent transport to the mitochondria that are driving the observed increase in folate?

Response

The transgene used in this work is the same as the previous studies. It is under the transcriptional control of a *S. cerevisiae* COX2 mitochondrial promoter previously demonstrated to be specifically expressed within the plant mitochondria, so we do not believe erroneous nuclear expression would be an issue. This is clarified within the main text (pg. 23) as follows:

(pg. 23)

Plasmids containing the GFP and luciferase reporter constructs were previously prepared and were previously shown to express exclusively within the mitochondria of *A. thaliana* (Plasmid maps are included in Supplementary Fig. 22)^{12,13,46}. The pDONR cox2p-RLuc and GFP constructs contain the promoter from *S. cerevisiae* COX2 and the reporter construct, without a terminator sequence. Previous studies have demonstrated that deletion of the untranslated terminator sequence did not affect expression of pDNA containing reporter constructs introduced within plant mitochondria^{47,48}.

Comment #23

Line 293. In the mt genotyping experiment, could the authors provide the primer sequences used as well as that of the donor plasmid? It is unclear if the authors used primers that would span the gaps of the HDR template and the mtgenome itself or if regions within the plasmid were used. If it is the latter case, could the amplified pcr product not originated from the delivered plasmid itself? Additionally, for plastid transformation, and HDR in mammalian cells, a linear donor template is preferred so it is surprising a circular construct is effective. Actual sequencing of an amplicon that spans the gaps of the HDR donor within the mtDNA would strongly supplement the claims made by the author here.

Response

The primers used and donor plasmid maps are now included in Supplementary Table 1 and Supplementary Fig. 22, respectively.

The primers used amplifies a PCR product spanning the junction between the HDR template and the mitochondrial genome and would only be amplified upon successful integration in the mitochondrial genome. The original figure in Fig. 6a showing primer binding sites was incorrect and has been amended.

Sequencing data has been added to Supporting Fig. 15, and the raw sequencing data has also been included as Supplementary Data 1 and 2.

The main text (pg. 16) has been amended as follows

(pg. 16)

DNA integration events were observed in all samples treated with SWNT-PM-CytKH9/pAtMTTF1 complexes, as indicated by positive genotyping PCR products for both the left and right homology arms (Fig. 6b). **These PCR products span the junctions between the mitochondrial genome and the homologous sequence from the donor plasmid, and thus would only be amplified in the case of successful recombination.** (Fig. 6b). In contrast, positive PCR products were detected faintly in only one sample out of six treated with pAtMTTF1 alone.

Fig. 6. Genotyping and phenotypic effects in *A. thaliana* upon infiltration with SWNT-PM-CytKH9/pAtMT pDNA complexes.

(a) Design of a plasmid DNA construct (pAtMTTF1) for integration into the mitochondrial genome of *A. thaliana*. *Sul1* encodes dihydropteroate synthase type-2, which was previously used as a selection marker for mitochondrial transformation³⁶.

Supplementary Fig. 15. Sequencing of the genotyping PCR products from the 5' and 3' arms of the integrated pAtMTTF1 construct. Sequencing reads spanning the junction between the mitochondrial genome and the homologous sequence from the donor plasmid for recombination on both arms confirmed integration at the expected positions within the mitochondrial genome. Primers 1 and 4 refer to the primers labelled in Fig. 6a. Raw sequence reads can be found in the Supplementary Data 1 and 2.

Supplementary Fig. 22. Plasmid maps of the reporter constructs pAtMTTF1, pDONR-Cox2Rluc, and pDONR-Cox2GFP.

Name	Sequence	Primer Number (Fig. 6a)
ATMTTF1-Up	ATCGAGATGCACATGGGATTGG	1
sul-F6	AGCGCAATCACCATCTCGGAAACC	2
GFP-F2	TGGTCCTGCTGGAGTTCGTGAC	3
ATMTTF1-Down	ATGGTTTGCACCTCCGTGTACTAG	4

Supplementary Table 1. List of PCR primers used for genotyping experiments in Fig. 6 and Supplementary Fig. 15.

Reviewer #3:

In the present paper the authors designed single-walled carbon nanotubes (SWCNT) for gene delivery in mitochondria. In particular they reported a detailed chemico-physical characterization of SWCNTs functionalised with peptides and plasmids. They employed two peptides to confer mitochondria targeting properties to SWCNT and increase DNA binding on the nanotubes. Then they checked the pDNA binding properties of this material before infiltrating them in plantlet of *A. thaliana*. They also proved that SWCNT-CA-pDNA enable expression of reporter genes in Arabidopsis and drive genetic integration through the homologous recombination of *sull* gene involved in the folate pathway

Nanomaterials for genetic engineering are highly appealing although a technological gap still limits their use in plant. The paper has the merit to provide a new strategy for SWCNT-mediated transformation in plant mitochondria. The chemical-physical characterization of the nanomaterials is solid while some flaws are in the description of the uptake mechanisms. The specificity of mitochondrial transformation is not ensured but the approach sounds interesting.

The significance and the novelty of the manuscript is suitable for publication on Nature communications. However, before publishing the following points need to be revised:

Abstract

Comment #1

Line 26: which method do they refer to?

Response

This refers to peptide-DNA mediated delivery and has been added to the abstract (pg. 2).

(pg. 2)

“Our previous studies demonstrated that plasmid DNA (pDNA) complexed with peptides containing the polycationic sequence (KH9) and the mitochondria targeting sequence (Cytcox) was capable of delivering DNA into mitochondria^{12,13}.”

Introduction

Comment #2

Line 81 - Information about Cytcox are not exhaustive and the reference of its targeting properties in plant is not appropriate. Similarly, KH9 details and relative references are missing

Response

Additional information regarding Cytcox and KH9 peptides have been added to both the introduction and results sections.

(pg. 4)

The Cytcox peptide contains the mitochondrial targeting presequence that has been previously characterized and used for mitochondrial pDNA delivery and expression within plant and animal cells²⁶. KH9 is a cationic peptide that facilitates electrostatic interactions with anionic pDNA for binding.”

(pg. 5)

Cytox Cys (MLSLRQSIRFFKC), abbreviated as Cyt in SWNT-PM-CytKH9, has been previously shown to direct the yeast cytochrome c oxidase subunit IV into the mitochondrial matrix and deliver pDNA for transient expression into mitochondria within plant and animal cells²⁶. KH9 (KHKHKHKHKHKHKHKHC) is a cationic peptide consisting of alternating lysine and histidine residues that facilitates electrostatic complexation with anionic pDNA to increase loading efficiency. Our previous studies using these peptides have shown that they are able to direct nucleic acid and protein cargoes to the mitochondria within intact plants^{12,13}.

Results

Comment #3

Lines 87-93 This sentence appears confused and should be rephrased

Response

This sentence has been rewritten and simplified as follows:

(pg. 5-6)

“The SWNT nanocarriers (NCs) were quantitatively and qualitatively evaluated for their physical and chemical properties. Field-emission scanning electron microscopy (FE-SEM) of SWNT-PM and SWNT-PM-CytKH9 (6 kV on Si) (Fig. 2a and b) show the typical bundled morphology of SWNTs approximately 200 nm to 2 μm in length, suggesting that the adsorption of the polymer layer and conjugation of the peptides did not significantly alter the physical dimensions of the SWNTs.”

Comment #4

Line 124. Please include a map of this plasmid

Response

A map of this plasmid has been added to Supplementary Fig. 22.

Supplementary Fig. 22. Plasmid maps of the reporter constructs pAtMTTF1, pDONR-Cox2luc, and pDONR-Cox2GFP.

Comment #5

Line 130. Looking at the graph in Fig. 2f (blue curve), the ratio SWNT-PM(KH9)/ pDNA (1 microgram/ 50 ng) seems not well estimated.

Response

The blue curve may have looked to be not well estimated due to the smearing of the DNA band in the gel that was not well depicted originally in Fig. 2e. Fig. 2e has been adjusted to show the band clearer. Furthermore, the uncropped versions of the gel are also now included in Supplementary Fig. 5. The revised versions are as follows:

Supplementary Fig. 5. Binding of pDNA to SWNT-PM-Peptide NCs.

Representative gel shift electrophoresis mobility assay used for quantification in Fig. 2e. The main unbound band at approximately 4kb was used for comparison in the binding abilities of the SWNT-PM-Peptide NCs.

Comment #6

Lines 134-137 What about the zeta potential of SWCNT-PM after plasmid binding? It should be reported.

Response

Zeta potentials of the respective SWNT/pDNA complexes have been added to the graph. We have also added the following information into the main text:

(pg. 8)

This was also reflected in the change in zeta potential upon complexation with pDNA, where SWNT-PM-KH9 saw the greatest change (-22.2 mV) when compared with SWNT-PM-CytKH9 (-4.8 mV), and SWNT-PM-Cytcox (+2.4 mV).

Fig. 2 (g) Zeta potentials of SWNT-PM-Peptide with different functional peptides in the absence and presence of pDNA. Data are represented as the mean \pm standard deviation values ($n = 3$).

Comment #7

Lines 173-176. FE-SEM images of infiltrated plants must be integrated with micrographs of non infiltrated plants treated with nanotubes and not infiltrated. FE-SEM are not totally convincing and could not discriminate clearly SWCNT-PM from naturally occurring granules available in the organelles. Therefore, the author must prove with additional techniques the mitochondrial localization. For instance, TEM images reported in literature could be much more useful than SEM to prove mitochondria uptake of SWNTs as reported in:

Ma X., et al 2021 ACS Nano. 2012;6(12):10486-10496. doi:10.1021/nm302457v

Battigelli et al., Nanoscale, 2013, 5, 9110-9117

- Alternatively, fluorescently -labelled nanotubes could be employed for mitochondrial colocalization analyses by confocal microscopy.

Response

Fluorescently-labelled SWNT-PM-CytKH9 using DyLight 488 were prepared and repeated for the infiltration experiments. Labelled SWNT-PM-CytKH9 were detected within the mitochondria, providing evidence for the uptake of the SWNT complexes into the mitochondria within the plants (Fig. 3a). Comparison between the non-infiltrated samples and infiltrated samples show that the SWNT complex does not readily enter the cell without vacuum infiltration (Fig. 3a).

Furthermore, confocal Raman microscopy showing mapping at the characteristic SWNT G-band peak (1590 cm^{-1}) of isolated mitochondria show strong colocalization of this signal with the mitochondria that are not observed in the samples without infiltration or the ones

infiltrated with SWNT-PM. This suggests that the Cytcox peptides are able to confer the ability of the SWNT to enter the mitochondria.

The results of these experiments have been summarized within the main text and Fig. 3 as follows:

To investigate the delivery capabilities of the SWNT NCs to plant mitochondria, we prepared fluorescently labelled SWNT-PM-CytKH9 using DyLight 488-conjugated KH9Cys and examined its localization within the root cells of *A. thaliana* upon vacuum/pressure infiltration at 0.08 MPa for 60 seconds. In the cells infiltrated with the labelled SWNT-PM-CytKH9, clear colocalization between DyLight 488-labelled SWNT-PM-CytKH9 and MitoTracker-stained mitochondria could be observed in the samples infiltrated with the labelled SWNT-PM-CytKH9 as well as localization of the labelled SWNT-PM-CytKH9 within the cytosol of the root cells (Fig. 3a, upper panels). Conversely, most of the labelled SWNT-PM-CytKH9 samples remained on the surface of the root without vacuum infiltration (Fig. 3a, lower panels).

Next, we analyzed isolated mitochondria from *A. thaliana* infiltrated with SWNT NCs by confocal Raman microscopy (Fig. 3b-d). Raman mapping of the G band peak at 1590 cm^{-1} showed clear colocalization of SWNT signal with the isolated mitochondria from seedlings infiltrated with SWNT-PM-KH9 and SWNT-PM-CytKH9 (Fig. 3b) that is not present in the samples infiltrated with SWNT-PM. Similar levels of colocalization was also observed in samples infiltrated with SWNT-PM-CytKH9 but those infiltrated with SWNT-PM-KH9 and SWNT-PM exhibited considerably less G-band signal, suggesting that the Cytcox peptide played a role in directing the SWNT NCs into the mitochondria. To further quantify the effect of the peptide conjugation, we collected Raman spectra over an area of isolated mitochondria treated with the SWNT NCs and the overlay of the averaged Raman spectra (Fig. 3c) showed that similar characteristic G and G' bands at 1590 cm^{-1} and 2640 cm^{-1} (Fig. 3c and d). Quantification of the normalized 1590 cm^{-1} (Fig. 3d) and 2640 cm^{-1} (Supplementary Fig. 8) peak heights showed that all three SWNT NCs were detected within the isolated mitochondria. In particular, the strongest Raman signals were detected from the samples infiltrated with SWNT-PM-CytKH9 with an approximately 10-fold increase in normalized G-band intensity relative to the samples infiltrated with SWNT-PM and SWNT-PM-KH9, suggesting that the dually functionalized SWNT was delivered most efficiently into the mitochondria (Fig. 3d and Supplementary Fig. 8). Taken together with the fluorescently-labelled SWNT results, these findings show that the SWNT NCs can be localized within plant mitochondria, with the Cytcox peptide conferring increased mitochondrial targeting.

Fig. 3. CLSM of fluorescently labelled SWNT NCs and Raman microscopy of mitochondria in *A. thaliana* upon introduction of SWNT NCs.

(a) Fluorescently-labelled (DyLight488) SWNT-PM-CytKH9 were detected within MitoTracker-labelled mitochondria and cytosol upon vacuum infiltration (upper panels) and on the surface of the root cells without vacuum infiltration (lower panels) when introduced into intact *A. thaliana* seedlings. White arrows indicate colocalization of the DyLight488-labelled SWNT-PM-CytKH9 and MitoTracker-labelled mitochondria. **(b)** Raman microscopy mapping of the 1490 cm^{-1} G-band peak taken from isolated mitochondria isolated from *A.*

thaliana 18 h after infiltration showed colocalization of SWNT signal with the isolated mitochondria in the samples infiltrated with SWNT-PM-CytKH9 and SWNT-PM-KH9. (c) Representative Raman spectra taken from mitochondria isolated from *A. thaliana* 18 h after infiltration show characteristic G (1590 cm^{-1}) and G' (2640 cm^{-1}) bands from the SWNTs. (d) Quantified normalized G-band peak (1590 cm^{-1}) intensities for isolated mitochondrial samples shown in (b). Relative values from 64 individual spectra ($n=64$) are shown for each individual sample. ns – not statistically significant, *** $P<0.001$, **** $P<0.0001$.

Comment #8

Moreover, the uptake analysis must be strengthened by quantitative data, for instance by reporting the number of SWCNT positive mitochondria vs the total observed in the micrographs.

Response

Quantification and comparison of uptake between the different types of SWNT NCs was performed by analysis of the Raman spectra. Mitochondria infiltrated with SWNT-PM-CytKH9 showed approximately 10-fold increase in normalized G-band intensity (1590 cm^{-1}) compared to the SWNT-PM and SWNT-PM-KH9 infiltrated samples. We have added this information within the main text (pg. 11) and Fig. 3d as follows:

(pg. 11)

To further quantify the effect of the peptide conjugation, we collected Raman spectra over an area of isolated mitochondria treated with the SWNT NCs and the overlay of the averaged Raman spectra (Fig. 3c) showed that similar characteristic G and G' bands at 1590 cm^{-1} and 2640 cm^{-1} (Fig. 3d). Quantification of the normalized 1590 cm^{-1} (Fig. 3d) and 2640 cm^{-1} (Supplementary Fig. 8) peak heights showed that all three SWNT NCs were detected within the isolated mitochondria. In particular, the strongest Raman signals were detected from the samples infiltrated with SWNT-PM-CytKH9 with an approximately 10-fold increase in normalized G-band intensity relative to the samples infiltrated with SWNT-PM and SWNT-PM-KH9, suggesting that the dually functionalized SWNT was delivered most efficiently into the mitochondria (Fig. 3c). Taken together with the fluorescently-labelled SWNT results, these findings show that the SWNT NCs can be localized within plant mitochondria, with the Cytcox peptide conferring increased mitochondrial targeting.

Fig. 3. CLSM of fluorescently labelled SWNT NCs and Raman microscopy of mitochondria in *A. thaliana* upon introduction of SWNT NCs.

(a) Fluorescently-labelled (DyLight488) SWNT-PM-CytKH9 were detected within MitoTracker-labelled mitochondria and cytosol upon vacuum infiltration (upper panels) and on the surface of the root cells without vacuum infiltration (lower panels) when introduced into intact *A. thaliana* seedlings. White arrows indicate colocalization of the DyLight488-labelled SWNT-PM-CytKH9 and MitoTracker-labelled mitochondria. **(b)** Raman microscopy mapping of the 1490 cm^{-1} G-band peak taken from isolated mitochondria isolated from *A. thaliana* 18 h after infiltration showed colocalization of SWNT signal with the isolated

mitochondria in the samples infiltrated with SWNT-PM-CytKH9 and SWNT-PM-KH9. (c) Representative Raman spectra taken from mitochondria isolated from *A. thaliana* 18 h after infiltration show characteristic G (1590 cm^{-1}) and G' (2640 cm^{-1}) bands from the SWNTs. (d) Quantified normalized G-band peak (1590 cm^{-1}) intensities for isolated mitochondrial samples shown in (b). Relative values from 64 individual spectra (n=64) are shown for each individual sample. ns – not statistically significant, ***P<0.001, ****P<0.0001.

Comment #9

Finally, the uncertainty to recognise SWNT from other organelle granules (as admitted by the authors at lines 178 -180) suggests to clearly state in the text that the SWCNT localization in other organelles cannot be excluded. The expression of nuclear promoters supports this.

Response

We have added the information within the text to clearly state that SWNT localization cannot be excluded from other organelles including the nucleus as shown the expression observed with the nuclear promoters with both the RLuc and GFP reporter constructs.

(pg. 14-15)

We note that the SWNT NCs likely had a degree of non-specificity and could localize within the nucleus in addition to the mitochondria as evidenced by the expression with the 35S promoter containing pDNA.

Conclusion

Comment #10

Apart the valorisation of the results, the conclusion must mention also the limit of nuclear expression that could be a problem for those applications that may require specific mitochondrial expression. The author should include these considerations for proper information.

Response

We have added statements into the conclusion regarding the considerations of the non-specific localization of the SWNT NCs that could limit applications into the conclusion. In the case of specific organelle gene expression, organelle-specific promoters can be used to ensure that expression is only observed in the target organelle.

(pg. 18)

Although the SWNT NCs also localized within the nucleus of the plant cells and could potentially limit their applications where mitochondrial-specific localization of the SWNT NCs are required, we believe sufficient specificity can be achieved by the tailoring the cargo such as by using organelle-specific promoters or translocation sequences.

Reviewers' Comments:

Reviewer #1:

Remarks to the Author:

The authors have made extensive, and I judge productive changes in response to my comments (and to the other reviewers as well). I find the manuscript much clearer now and the extensive additions make do clarify the approaches, results and conclusions, and note the potential limitations.

The two outstanding question that remain for me are more high-level and cannot be addressed by revision:

- 1) Is this increase in efficiency of mitochondrial transformation significant to justify publication – I find this hard to judge but overall I believe it is and would be of interest to a wide variety of plant biologists.
- 2) The enhanced growth phenotype – still surprised by this – I would have thought a standard transformation of nuclear genome of with a construct that would produce thee 'same' protein targeted to mitochondria would be a good verification of this results. This has not been carried out. However, referencing previous reports that show folic acid affects root branching and auxin to some way help explain this phenotype.

Overall I believe that the report has made a novel and significant contribution that would be of high interest to a number of plant biologists.

Reviewer #2:

Remarks to the Author:

The authors have addressed my prior concerns in a satisfactory manner and the manuscript is suitable for publication.

Reviewer #3:

Remarks to the Author:

The authors provided a sufficient amount of work to address the reviewers concerns. I support the publication of this paper as it introduces important novelty in plant nanobiotechnology. However, the confocal images reported to prove the colocalization of SWNT show some flaws and can be improved. Below my specific comments:

- Confocal imaging and colocalization are not convincingly described and displayed. Important details about the method are not described in materials and methods. The cell wall in infiltrated plants is not visible. Did the authors use a cell wall staining? If so, why the cells are not in the focal plane? At the magnification reported it is impossible to claim a real colocalization. Could the authors provide close up images of colocalization?
- If this experiment was conceived to prove the specific targeting properties of the nanomaterials, I would say that not-infiltrate plants are not a good control. In my opinion, free labelled-peptides or even better unfuctionalized fluorescently labelled-SWNT upon infiltration would provide more useful information than the current ones.
- Please, include in Figure 2 caption what the white arrows represent.
- Revise some typos throughout the manuscript

Responses to the Reviewers' Comments

We thank the reviewers for their additional comments to further improve our manuscript. Our replies to their individual comments are colored in red with direct changes to the main text highlighted in yellow. Relevant figures have also been reproduced.

Reviewer #1

The authors have made extensive, and I judge productive changes in response to my comments (and to the other reviewers as well). I find the manuscript much clearer now and the extensive additions make do clarify the approaches, results and conclusions, and note the potential limitations.

The two outstanding question that remain for me are more high-level and cannot be addressed by revision:

Comment #1

Is this increase in efficiency of mitochondrial transformation significant to justify publication – I find this hard to judge but overall I believe it is and would be of interest to a wide variety of plant biologists.

Response

This manuscript reports for the first time that a SWNT-based delivery system that can be used to deliver pDNA into the mitochondria of intact plants with great potential flexibility for adapting to other organelle targets and for different cargo.

This technique also demonstrated a significant increase in mitochondrial transformation efficiency, which suggests its utility in further applications such as metabolic engineering of plant mitochondria using transient expression or stable transformation, as shown by our experiments with the *SUL1* construct. Furthermore, this technique can be used to facilitate the development of a mitochondrial selection marker system that would enable stable and inheritable transformation of plant mitochondria for the engineering of important agronomic traits.

Comment #2

The enhanced growth phenotype – still surprised by this – I would have thought a standard transformation of nuclear genome of with a construct that would produce thee ‘same’ protein targeted to mitochondria would be a good verification of this results. This has not been carried out. However, referencing previous reports that show folic acid affects root branching and auxin to some way help explain this phenotype.

Response

The experiments comparing the negative control pDNA without the *SUL1* insert (Supplementary Fig. 20) also provide strong evidence that *SUL1* expression directly contributed to the observed increase in root growth.

Also, as highlighted in the manuscript, folate end products are able to affect root growth and development within *A. thaliana* through auxin signaling and other developmental pathways. The expression of *SUL1* upon integration into the mitochondrial genome displayed elevated folate levels in roots (quantified in Supplementary Fig 21), and likely played a role in the observed increase in root growth.

Supplementary Fig. 20. Effect on *A. thaliana* root growth by infiltration of SWNT-PM-CytKH9/pDNA with and without the *SUL1* insert.

A. thaliana root growth after infiltration with SWNT-PM-CytKH9 complexed with pAtMTTF1 pDNA containing the *SUL1* insert (+*SUL1*) and the negative control with the -insert removed (-*SUL1*) relative to their respective pDNA only control. Significant differences were observed between the relative root areas (n=18 seedlings) on Day 3, 5, and 7. The bars displayed represent the mean and the error bars represent the standard deviation.

Supplementary Fig. 21. Photosynthetic performance, mitochondrial respiration activities, and intracellular folate concentrations of *A. thaliana* infiltrated with SWNT NCs and its pDNA complex. (b) Quantification of chlorophyll fluorescence (F_v/F_m) of samples from (a) across 2 plates per condition containing a minimum of 10 leaf areas each using the FluorCam software (Version 7) showed no significant difference between the measured samples. Average values of F_v/F_m ranging from 0.75-0.80 for all sample suggest low levels of stress in plants even after SWNT NC treatment. **(c)** Mitochondrial respiration as quantified by ATP generation from isolated mitochondria from *A. thaliana* showed no significant differences in respiration or mitochondrial stress upon infiltration with SWNT NCs and its complex or DNA only. **(d)** Folic acid quantification of *A. thaliana* infiltrated with SWNT NCs or DNA only show a significant increase in folic acid levels at Days 1 and Day 3 post-infiltration for SWNT-PM-CytKH9/pDNA (+*SUL1*) treated samples relative to the pDNA only and SWNT-PM-CytKH9 samples and at Day 1 relative to SWNT-PM-

CytKH9/pDNA (-*SUL1*). For all plots, the bars displayed represent the mean and the error bars represent the standard deviation.

Overall I believe that the report has made a novel and significant contribution that would be of high interest to a number of plant biologists.

We sincerely appreciate Reviewer #1 for their review and encouraging comments regarding our manuscript. Your feedback and comments helped to significantly improve and refine our manuscript.

Reviewer #2

The authors have addressed my prior concerns in a satisfactory manner and the manuscript is suitable for publication.

We would like to thank Reviewer #2's time and effort in reviewing our manuscript. Your comprehensive analysis helped to considerably improve our manuscript.

Reviewer #3

The authors provided a sufficient amount of work to address the reviewers concerns. I support the publication of this paper as it introduces important novelty in plant nanobiotechnology. However, the confocal images reported to prove the colocalization of SWNT show some flaws and can be improved.

Below my specific comments:

Comment #1

Confocal imaging and colocalization are not convincingly described and displayed. Important details about the method are not described in materials and methods. The cell wall in infiltrated plants is not visible. Did the authors use a cell wall staining? If so, why the cells are not in the focal plane? At the magnification reported it is impossible to claim a real colocalization. Could the authors provide close up images of colocalization?

Response

We apologize for the lack of details regarding description of the materials and methods for confocal experiments and colocalization experiments, we have included additional information within the materials and methods as follows:

Seedlings vacuum infiltrated with the respective SWNT NCs were washed with dH₂O prior to staining for CLSM analysis (18 h incubation post infiltration for expression and 3 h for labelled SWNT). Mitochondria were stained using 100 nM MitoTracker Red CMXRos (Thermo Fisher, USA) for 1 h at room temperature and washed with dH₂O three times to remove excess dye before imaging⁵¹. Intact seedlings were placed on a microscope slide and suspended in dH₂O for CLSM measurements. Labelled mitochondria within the roots of *A. thaliana* were imaged using an excitation and emission (Ex/Em) wavelengths of 555/580-610 nm (for CMXRos). The intracellular localization of the GFP expression and DyLight488 were imaged at 488/500-540 nm and 488/500-540 nm, respectively.

Raman microscopy analyses of isolated mitochondria samples and the SWNT NCs were performed using a JASCO NRS-4500 Raman microscope (JASCO, Japan). Nanoparticle characterization was performed using aqueous samples at 100x on a cover glass with a green laser at 532 nm with an integration time of six seconds per spectra that were collected over an area of 50 μm x 40 μm. Isolated mitochondrial samples were prepared similarly for the western blot analyses using freshly prepared and isolated mitochondria that were imaged as aqueous samples on a glass slide immediately after isolation using the same conditions. Raman mapping studies were performed using a Raman Touch confocal Raman microscope (Nanophoton Corp., Japan) using a green laser at 532 nm collecting in line mapping mode.

For the imaged samples, the mitochondria were stained with MitoTracker CMXRos, which had a tendency to nonspecifically stain the cell membranes as well, as seen in many samples (the cell walls were not directly stained).

We have replaced the previous images with close-up images of the colocalized mitochondria to better illustrate the colocalization of the fluorescent label and the labelled

mitochondria. Control samples with only free labelled-peptides have also been included in the bottom panels as suggested. The DIC images are also now shown as a separate panel for clarity. The main text has also been updated accordingly:

Conversely, most of the labelled SWNT-PM-CytKH9 samples remained on the surface of the root without vacuum infiltration (Fig. 3a, middle panels) and minimal fluorescence was observed for seedlings infiltrated with the free DyLight488-labelled KH9 (Fig. 3a, lower panels).

Fig. 3. CLSM of fluorescently labelled SWNT NCs and Raman microscopy of mitochondria in *A. thaliana* upon introduction of SWNT NCs.

(a) Fluorescently-labelled (DyLight488) SWNT-PM-CytKH9 were detected within MitoTracker-labelled mitochondria and cytosol upon vacuum infiltration (upper panels) and on the surface of the root cells without vacuum infiltration (middle panels) when introduced into intact *A. thaliana* seedlings. White arrows indicate colocalization of the DyLight488-labelled SWNT-PM-CytKH9 and MitoTracker-labelled mitochondria. Minimal fluorescence was observed for the *A. thaliana* seedlings infiltrated with the labelled peptide only (bottom panels). Scale bars represent 10 μm .

Comment #2

- If this experiment was conceived to prove the specific targeting properties of the nanomaterials, I would say that not-infiltrate plants are not a good control. In my opinion, free labelled-peptides or even better unfuctionalized fluorescently labelled-SWNT upon

infiltration would provide more useful information than the current ones.

Response

This experiment was conceived to provide additional confirmation of the localization of the SWNT-PM-CytKH9 within the mitochondria in addition to the Raman mapping (Fig 3b) and Raman spectral quantification (Fig. 3c-d) experiments that provide strong evidence of the colocalization of the delivered SWNT and plant mitochondria.

For the direct comparison of the targeting properties of the different peptides, we feel the Raman mapping and the quantification of the unique SWNT G bands, and to a lesser extent the transfection experiments involving expression of the RLuc pDNA using the different SWNT NCs, provide a better assessment of their differences in targeting efficiency due to the potential cleavage of the fluorescent signal with the labelled-peptides and non-specificity of the dye that may affect such comparisons with confocal microscopy.

As suggested, we have also included images of *A. thaliana* infiltrated with free DyLight 488-labelled KH9 used to label the SWNT-PM-CytKH9 and have included this in Fig. 3a (bottom panels).

Fig. 3. CLSM of fluorescently labelled SWNT NCs and Raman microscopy of mitochondria in *A. thaliana* upon introduction of SWNT NCs.

(a) Fluorescently labelled (DyLight488) SWNT-PM-CytKH9 were detected within MitoTracker-labelled mitochondria and cytosol upon vacuum infiltration (upper panels) and on the surface of the root cells without vacuum infiltration (middle panels) when introduced into intact *A. thaliana* seedlings. White arrows indicate colocalization of the DyLight488-

labelled SWNT-PM-CytKH9 and MitoTracker-labelled mitochondria. Minimal fluorescence was observed for the *A. thaliana* seedlings infiltrated with the labelled peptide only (bottom panels). Scale bars represent 10 μm .

Comment #3

Please, include in Figure 2 caption what the white arrows represent.

Response

The following has been added into the Figure 2 caption:

White arrows indicate complexed pDNA on the surface of the SWNT.

Comment #4

Revise some typos throughout the manuscript

Response

Several typos have been corrected in the manuscript.

“Were” was corrected to “was” on page 9.

“Was” was corrected to “were” on page 10.

“That” was deleted on page 10.

“upon” was corrected to “using” on page 11.

“detectable” was changed to “appreciable” on page 13.

“as well as organelle-targeting peptides such as Cytcox” was deleted on page 15.

“Then” was deleted on page 20.

We sincerely appreciate Reviewer #3 for the positive review of our manuscript. Your comments helped significantly in revising and improving our manuscript.

Reviewers' Comments:

Reviewer #3:

Remarks to the Author:

The authors have properly addressed my last concerns. In my opinion the manuscript now is suitable for publication.

Responses to the Reviewers' Comments

We thank the reviewers for their additional comments to further improve our manuscript.

Reviewer #3

The authors have properly addressed my last concerns. In my opinion the manuscript now is suitable for publication.

We sincerely thank Reviewer #3 for their positive and detailed comments to improve our manuscript.